# Tree Mover's Distance: Bridging Graph Metrics and Stability of Graph Neural Networks

**Ching-Yao Chuang**
MIT CSAIL
cychuang@mit.edu

**Stefanie Jegelka**
MIT CSAIL
stefje@mit.edu

## Abstract

Understanding generalization and robustness of machine learning models fundamentally relies on assuming an appropriate metric on the data space. Identifying such a metric is particularly challenging for non-Euclidean data such as graphs. Here, we propose a pseudometric for attributed graphs, the Tree Mover's Distance (TMD), and study its relation to generalization. Via a hierarchical optimal transport problem, TMD reflects the local distribution of node attributes as well as the distribution of local computation trees, which are known to be decisive for the learning behavior of graph neural networks (GNNs). First, we show that TMD captures properties relevant to graph classification: a simple TMD-SVM performs competitively with standard GNNs. Second, we relate TMD to generalization of GNNs under distribution shifts, and show that it correlates well with performance drop under such shifts. The code is available at https://github.com/chingyaoc/TMD.

## 1 Introduction

Understanding generalization under distribution shifts – theoretically and empirically – relies on an appropriate measure of divergence between data distributions. This, in turn, typically demands a metric on the data space that indicates what kinds of data points are "close" to the training data. While such metrics may be readily available for Euclidean spaces, they are more challenging to determine for non-Euclidean data spaces such as graphs with attributes in $\mathbb{R}^p$, which underlie graph learning methods. In this work, we study the question of a suitable metric for message passing graph neural networks (GNNs).

An "ideal" metric for studying input perturbations captures the invariances and inductive biases of the model we are examining. For instance, since graph isomorphism is a difficult problem [18], this metric is expected to be a pseudometric, i.e., will fail to distinguish certain graphs. These "failures" should be aligned with the GNNs' invariances. Moreover, several recent works highlight the importance of local structures – computation trees resulting from unrolling the message passing process – for the approximation power of GNNs and their inability to distinguish certain pairs of graphs [2, 20, 34, 55]. Works on out-of-distribution generalization of GNNs mostly focus on specific types of instances where a trained model may fail miserably [58, 54], without specifying the behavior for gradual shifts in distributions. Yehudai et al. [58] show that a sufficiently large, unrestricted GNN may predict arbitrarily on previously unseen computation trees. In practice, we may observe a more gradual change, depending on the magnitude of change of the tree, in terms of both structure and node attributes, and the capacity of the aggregation function. In summary, we desire a metric that reflects the structure of computation trees and the distribution of node attributes *within* trees. Many distances and kernels between graphs have been proposed [7], several based on local structures [21], and some using structure and attributes [44]. Closest to our ideal is the *Wasserstein Weisfeiler-Leman* pseudometric [46], which computes an optimal transport distance between node embeddings of two graphs. The embeddings are computed via message passing, which aligns with the computation of GNNs, but loses structural information within trees.

36th Conference on Neural Information Processing Systems (NeurIPS 2022).

Hence, we propose the *Tree Mover's Distance (TMD)*, a pseudometric on attributed graphs that considers both the tree structure and local distribution of attributes. It achieves this via a hierarchical optimal transport problem that defines distances between trees. First, we observe that the TMD captures properties that capture relationships between graphs and common labels: a simple SVM based on TMD performs competitively with standard GNNs and graph kernels on graph classification benchmarks. Second, we relate TMD to the performance of GNNs under input perturbations. We determine a Lipschitz constant of GNNs with respect to TMD, which enables a bound on their target risk under domain shifts, i.e., distribution shifts between training and test data. This bound uses the metric in two ways: to measure the distribution shift, and to measure perturbation robustness via the Lipschitz constant. Empirically, we observe that the TMD correlates well with the performance of GNNs under distribution shifts, also when compared to other distances. We hence hope that this work inspires further empirical and theoretical work on tightening the understanding of the performance of GNNs under distribution shifts.

In short, this work makes the following contributions:
- We propose a new graph metric via hierarchical optimal transport between computation trees of graphs, which, in an SVM, leads to a competitive graph learning method;
- We bound the Lipschitz constant of message-passing GNNs with respect to TMD, which allows to quantify stability and generalization;
- We develop a generalization bound for GNNs under distribution shifts that correlates well with empirical behavior.

## 2 Related Works

**Graph Metrics and Kernels**   Measuring distances between graphs has been a long-standing goal in data analysis. However, proper *metrics* that distinguish non-isomorphic graphs in polynomial time are not known. For instance, Vayer et al. [49] propose a graph metric by fusing Wassestein distance [50] and Gromov-Wasserstein distance [32]. Similar to classic graph metrics [9, 40], the proposed metric requires approximation. Closely related, graph kernels [52, 7] have gained attention. Most graph kernels lie in the framework of $\mathcal{R}$-convolutional [21], and measure similarity by comparing substructures. Many $\mathcal{R}$-convolutional kernels have limited expressive power and sometimes struggle to handle continuously attributed graphs [46]. In comparison, TMD is as powerful as the WL graph isomorphism test [53] while accommodating graphs with high dimensional node attributes. Importantly, TMD captures the stability and generalization of message-passing GNNs [27, 55].

**Stability and Generalization of Graph Neural Networks**   A number of existing works study stability of GNNs to input perturbations. Spectral GNNs are known to be stable to certain perturbations, e.g., size, if the overall structure is preserved [19, 25, 26, 19]. For message passing GNNs, Yehudai et al. [57] study perturbations of graph size, and demonstrate the importance of local computation trees. Xu et al. [54] study how the out-of-distribution behavior of aggregation functions may affect the GNN's prediction. These studies motivate to include both computation trees and inputs to aggregation functions in the TMD. Finally, a number of works study within-distribution generalization [15, 20, 28, 42].

## 3 Background on Optimal Transport

We begin with a brief introduction to Optimal Transport (OT) and earth mover's distance. The earth mover's distance, also known as Wasserstein distance, is a distance function defined via the transportation cost between two distributions. Let $X = \{x_i\}_{i=1}^m$ and $Y = \{y_j\}_{j=1}^m$ be two multisets of $m$ elements each. Let $C \in \mathbb{R}^{m \times m}$ be the transportation cost for each pair: $C_{ij} = d(x_i, y_j)$ where $d$ is the distance between $x_i$ and $y_j$. The earth mover's distance solves the following OT problem:

$$\text{OT}_d^*(X, Y) := \min_{\gamma \in \Gamma(X,Y)} \langle C, \gamma \rangle / m, \quad \Gamma(X, Y) = \{\gamma \in \mathbb{R}_+^{m \times m} \mid \gamma \mathbb{1}_m = \gamma^\top \mathbb{1}_m = \mathbb{1}_m\}, \quad (1)$$

where $\Gamma$ is the set of *transportation plans* that satisfies the flow constrain $\gamma \mathbb{1}_m = \gamma^\top \mathbb{1}_m = \mathbb{1}_m$. In this work, we adopt the *unnormalized* version of the earth's mover distance:

$$\text{OT}_d(X, Y) := \min_{\gamma \in \Gamma(X,Y)} \langle C, \gamma \rangle = m \cdot \text{OT}_d^*(X, Y). \quad (2)$$

Comparing to classic OT, unnormalized OT preserves the size information of multisets $X$ and $Y$.

Figure 1: **Tree Distance via Hierarchical OT.** The weight $w(\cdot)$ is set to 1 to simplify visualization. The distance between two trees can be decomposed into (a) the distance between roots and (b) the OT cost between subtrees, where the cost function of OT is again the distance between two trees. This formulates a *hierarchical* OT problem, where solving the OT between trees requires solving the OT between subtrees.

## 4 Tree Mover's Distance: Optimal Transport on Graphs

Next, we introduce *tree mover's distance*, a new distance for graphs. Let $G = (V, E)$ denote a graph with sets of nodes $V$ and edges $E$. The graph may have node features $x_v \in \mathbb{R}^p$ for $v \in V$. If not, we simply set the node feature to a scalar $x_v = 1$ for all $v \in V$.

Local structures of graphs are characterized by *computation trees* [23]. In particular, computation trees are constructed by connecting adjacent nodes recursively.

**Definition 1** (Computation Trees). *Given a graph $G = (V, E)$, let $T_v^1 = v$, and let $T_v^L$ be the depth-L computation tree of node $v$ constructed by connecting the neighbors of the leaf nodes of $T_v^{L-1}$ to the tree. The multiset of depth-L computation trees defined by $G$ is denoted by $\mathcal{T}_G^L := \{T_v^L\}_{v \in V}$.*

Figure 2 illustrates an example of constructing computation trees. Computation trees, also referred to as subtree patterns, have been central in graph analysis [38, 53] and graph kernels [39, 44]. Intuitively, the computation tree of a node encodes local structure by appending the neighbors to the tree in each level. If depth-$L$ computation trees are the same for two nodes, they share similar neighborhoods up to $L$ steps away. Therefore, an intuitive way to compare two graphs is by measuring the difference of their nodes' computation trees [44, 53]. In this work, we adopt optimal transport, a natural way to compute the distance between two sets of objects with, importantly, an underlying geometry. We will begin by defining the transportation cost between two computation trees. The cost then gives rise to the tree mover's distance, an extension of earth mover's distance to multisets of trees.

### 4.1 Distance between Trees via Hierarchical OT

Let $T = (V, E, r)$ denote a rooted tree. We further let $\mathcal{T}_v$ be the multiset of computation trees of the node $v$ which consists of trees that root at the descendants of $v$. Determining whether two trees are similar requires iteratively examining whether the subtrees in each level are similar. For instance, two trees $T_a$ and $T_b$ with roots $r_a$ and $r_b$ are the same if $x_{r_a} = x_{r_b}$ and $\mathcal{T}_{r_a} = \mathcal{T}_{r_b}$. This motivates us to define the distance between two trees by recursively computing the optimal transportation cost between their subtrees. Nevertheless, the number of subtrees could be different for $r_a$ and $r_b$, i.e., $|\mathcal{T}_{r_a}| \neq |\mathcal{T}_{r_b}|$ (see Figure 1 left). To compute the OT between sets with different sizes, unbalanced OT [11, 41] or partial OT [10] are usually adopted. Inspired by [10], we augment the smaller set with *blank trees*.

**Definition 2** (Blank Tree). *A blank tree $T_\emptyset$ is a tree (graph) that contains a single node and no edge, where the node feature is the zero vector $\mathbb{0}_p \in \mathbb{R}^p$, and $T_\emptyset^n$ denotes a multiset of $n$ blank trees.*

**Definition 3** (Blank Tree Augmentation). *Given two multisets of trees $\mathcal{T}_u, \mathcal{T}_v$, define $\rho$ to be a function that augments a pair of trees with blank trees as follows:*

$$\rho : (\mathcal{T}_v, \mathcal{T}_u) \mapsto \left( \mathcal{T}_v \bigcup T_\emptyset^{\max(|\mathcal{T}_u| - |\mathcal{T}_v|, 0)}, \mathcal{T}_u \bigcup T_\emptyset^{\max(|\mathcal{T}_v| - |\mathcal{T}_u|, 0)} \right).$$

If $|\mathcal{T}_v| < |\mathcal{T}_u|$, $\rho$ augments $\mathcal{T}_v$ by $|\mathcal{T}_u| - |\mathcal{T}_v|$ blank trees to make the two multisets contain the same number of trees, and hence allows to define a transportation cost between two multisets of trees with different sizes. In particular, the transportation costs of additional trees are simply the distance to the blank trees. The distance between a tree and a blank tree can be interpreted as the *norm of the tree*. In our case, the blank tree can be viewed as the origin, as it is a tree with the simplest structure and zero feature vector. Equipped with $\rho$, we define the distance between two rooted trees as follows.

**Definition 4** (Tree Distance). *The distance between two trees $T_a, T_b$ is defined recursively as*

$$\text{TD}_w(T_a, T_b) := \begin{cases} \|x_{r_a} - x_{r_b}\| + w(L) \cdot \text{OT}_{\text{TD}_w}(\rho(\mathcal{T}_{r_a}, \mathcal{T}_{r_b})) & \text{if } L > 1 \\ \|x_{r_a} - x_{r_b}\| & \text{otherwise}, \end{cases}$$

*where $L = \max(\text{Depth}(T_a), \text{Depth}(T_b))$ and $w : \mathbb{N} \to \mathbb{R}^+$ is a depth-dependent weighting function.*

Here, $\text{OT}_{\text{TD}_w}$ is the OT distance defined in (2) with $\text{TD}_w$ as the metric. Figure 1 gives an illustration of computing tree distances. The tree distance $\text{TD}_w(T_a, T_b)$ aims to optimally align two trees $T_a$ and $T_b$ by recursively comparing their roots and subtrees. Calculating $\text{TD}_w(T_a, T_b)$ requires calculating the OT between augmented subtrees $\rho(\mathcal{T}_{r_a}, \mathcal{T}_{r_b})$, where the cost function of OT is $\text{TD}_w$ again. This formulates a *hierarchical* optimal transport problem: the distance of two trees is defined via the distances of subtrees, where the importance of each level is determined by the weight $w(\cdot)$. Increasing $w(\cdot)$ upweights the effect of nodes in the subtrees. While the weights may be arbitrary, we found that for many applications, using a single weight for all depths yields good empirical performance, as section 4.3 shows. The role of weights will be more significant when we use TMD to bound the stability of GNNs.

## 4.2 From Tree Distance to Graph Distance

Next, we extend the distance between trees to a distance between graphs. By leveraging the tree distance $\text{TD}_w(\cdot, \cdot)$, we introduce the *tree mover's distance (TMD)*, a distance for graphs, by calculating the optimal transportation cost between the graphs' computation trees.

**Definition 5** (**Tree Mover's Distance**). *Given two graphs $G_a, G_b$ and $w, L \geq 0$, the tree mover's distance between $G_a$ and $G_b$ is defined as*

$$\text{TMD}_w^L(G_a, G_b) = \text{OT}_{\text{TD}_w}(\rho(\mathcal{T}_{G_a}^L, \mathcal{T}_{G_b}^L)),$$

*where $\mathcal{T}_{G_a}^L$ and $\mathcal{T}_{G_b}^L$ are multisets of the depth-$L$ computation trees of graphs $G_a$ and $G_b$, respectively.*

Figure 2 illustrates the computation of TMD. Intuitively, TMD is the minimum cost required to transport node-wise computation trees from one graph to another. The blank tree augmentation $\rho$ is again adopted to handle graphs with different numbers of nodes. The next theorem shows that TMD is a pseudometric on attributed graphs.

**Theorem 6** (Pseudometric). *The tree mover's distance $\text{TMD}_w^L$ is a pseudometric for finite $L > 0$.*

In particular, the tree mover's distance satisfies (1) $\text{TMD}_w^L(G_a, G_a) = 0$, (2) $\text{TMD}_w^L(G_a, G_b) = \text{TMD}_w^L(G_b, G_a)$, and (3) $\text{TMD}_w^L(G_a, G_b) \leq \text{TMD}_w^L(G_a, G_c) + \text{TMD}_w^L(G_c, G_b)$ for any graphs $G_a, G_b, G_c$. However, in some cases, the distance $\text{TMD}_w^L(G_a, G_b)$ can be zero even if $G_a \neq G_b$. This is reasonable, as computing graph isomorphism is not known to be solvable in polynomial time. Nevertheless, TMD can provably distinguish graphs that are identifiable by the (1-dimensional) Weisfeiler-Leman graph isomorphism test [53].

**Theorem 7** (Discriminative Power of TMD). *If two graphs $G_a$, $G_b$ are determined to be non-isomorphic in WL iteration $L$ and $w(l) > 0$ for all $0 < l \leq L + 1$, then $\text{TMD}_w^{L+1}(G_a, G_b) > 0$.*

The unnormalized OT and blank tree augmentation are essential to prove Theorem 16. The tree mover's distance can be exactly computed by solving optimal transport. In addition, TMD remains highly expressive on graphs with high dimensional continuous attributes, where most $\mathcal{R}$-convolutional graph kernels struggle [46]. The discriminative power of TMD can be further strengthened by augmenting node attributes e.g. with positional encodings [16, 29].

The OT cost between node representations in different graphs is reminiscent of the recently proposed Wasserstein WL (WWL) [46]. WWL uses the distance of node embeddings as a ground metric, where the node embeddings are computed via $L$ iterations of message passing with average-aggregation; in contrast, TMD computes a distance that aligns the trees, retaining more structural information. Even though an aggregation with nonlinearities can retain tree isomorphism information [55], similar to the hashing applied in the WL test (WWL uses only linear aggregations), the hierarchical OT is a more direct graded distance measure of trees. Vayer et al. [49] define a metric that uses a Gromov-Wasserstein distance [32] between nodes, but need to approximate the GW computation.

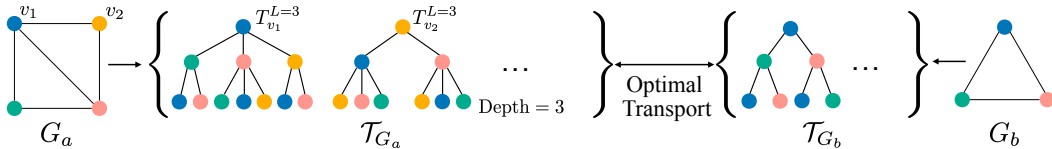

Figure 2: **Illustration of Computation Trees and Tree Mover's Distance.** The computation trees of nodes are constructed by iteratively connecting the neighbors to the trees, and each graph will define a multiset of node-wise computation trees. Tree mover's distance is then defined as the optimal transport cost between the computation trees of two graphs.

|          | MUTAG    | PTC      | PROTEINS | NCI1     | NCI109    | BZR      | COX2     |
|----------|----------|----------|----------|----------|-----------|----------|----------|
| TMD L=1  | 89.4±5.5 | 65.3±5.8 | 73.9±2.8 | 68.3±2.0 | 69.5±1.6  | 83.8±7.2 | 77.8±5.0 |
| TMD L=2  | 90.0±5.7 | 67.4±7.7 | 74.8±2.8 | 80.8±1.8 | 78.9±2.3  | 84.5±6.9 | **79.1±5.2** |
| TMD L=3  | 91.1±5.4 | **68.5±6.1** | 74.6±2.6 | 83.3±1.1 | 82.3±2.5  | **85.5±6.2** | 78.5±5.9 |
| TMD L=4  | **92.2±6.0** | 66.5±7.1 | 75.2±2.3 | 84.8±1.2 | **82.8±2.1** | 84.5±6.4 | 76.1±6.1 |
| WWL [46] | 87.3±1.5 | 66.3±1.2 | 74.3±0.6 | 86.1±0.3 | -         | 84.4±2.0 | 78.3±0.5 |
| FGW [49] | 88.4±5.6 | 65.3±7.9 | 74.5±2.7 | **86.4±1.6** | -     | 85.1±4.2 | 77.2±4.9 |
| WL [44]  | 90.4±5.7 | 59.9±4.3 | 75.0±3.1 | 86.0±1.8 | 82.46±0.2 | N/A      | N/A      |
| R&G [39] | 85.7±0.4 | 58.5±0.9 | 70.7±0.4 | 61.9±0.3 | 61.7±0.2  | N/A      | N/A      |
| GIN [55] | 89.4±5.6 | 64.6±7.0 | **76.2±2.8** | 82.7±1.7 | 82.2±1.6 | 83.5±6.0 | 79.0±5.3 |
| GCN [27] | 85.6±5.8 | 64.2±4.3 | 76.0±3.2 | 80.2±2.0 | -         | 84.6±5.9 | 77.1±4.7 |

Table 1: **Classification on TU Dataset.** TMD outperforms or matches the state-of-the-art graph kernels or GNNs. Note that WL [44] and R&G [39] are not applicable to continuously attributed graphs such as BZR and COX2.

**Numerical Computation with Dynamic Programming**   One can numerically compute TMD with dynamic programming. Starting with pair-wise distances ($\text{TMD}_w^{L=1}$) between node features across the two graphs, we then iteratively compute $\text{TMD}_w^{L=k}$ between depth-$k$ computation trees from $k = 2$ to $L$ according to Definition 4. Let $D$ be the maximum degree of a node in the two graphs and $\tau(m)$ be the complexity of computing OT between sets of cardinality $m$. In each level, we have to perform OT of sets contain at most $D$ elements for $N$ nodes. Including the last OT between nodes of graph, the overall time complexity of computing $\text{TMD}_w^L$ is $\mathcal{O}(\tau(N) + LN\tau(D))$. The time complexity for exact OT by solving linear programming is $\tau(m) = \mathcal{O}(m^3 \log(m))$ [17]. One can use faster approximation of OT, e.g., near linear time complexity [1], but we use exact OT throughout all the experiments, implemented with the POT library [17].

## 4.3   Experiments

We verify whether the tree mover's distance aligns with the labels of graphs in graph classification tasks: the TUDatasets [35], which contain graphs with discrete node attributes (MUTAG, PTC-MR, PROTEINS, NCI1, NCI109) and graphs with continuous node attributes (BZR, COX2). Specifically, we run a support vector classifier (C=1) with indefinite kernel $e^{-\gamma \times \text{TMD}(\cdot, \cdot)}$, which can be viewed as a noisy observation of the true positive semidefinite kernel [31]. The $\gamma$ is selected via cross-validation from $\{0.01, 0.05, 0.1\}$ and the weights $w(\cdot)$ are set to 0.5 for all depths. For comparison, we use graph kernels based on graph subtrees: Ramon & Gärtner kernel [39], WL subtree kernel [44]; two widely-adopted GNNs: graph isomorphism network (GIN) [55], graph convolutional networks (GCN) [27]; and the recently proposed graph metrics FGW [49] and WWL [46]. Table 1 reports the mean and standard deviation over 10 independent trials with 90%/10% train-test split. The performances of the baselines are taken from the original papers. TMD outperforms or matches the performances of state-of-the-art GNNs, graph kernels, and metrics, implying that it captures meaningful structural properties of graphs. Appendix C shows further graph clustering and t-SNE visualization [48] results.

**Computation Complexity** To examine the computation complexity, we compare the runtime between Weisfeiler-Lehman (WWL) kernels, TMD, and a parallel version of TMD on DD and NCI1 datasets [35], where DD contains large graphs (avg. #node: 284.32, avg. #edge: 715.66) and NCI1 contains small graphs (avg. #node: 29.87, avg. #rdge: 32.30). Here, we additionally consider a parallel version of TMD, where the tree OTs in each level are executed simultaneously with 3 processes. The average runtime over 200 pairs is shown in Table 2. Note that the

|      | WWL  | TMD   | TMD Parallel |
|------|------|-------|--------------|
| DD   | 7.92 | 32.07 | 24.44        |
| NCI1 | 0.11 | 0.34  | 0.81         |

Table 2: **Runtime Comparison.** The average runtime (sec/pair) of TMD is much faster than the (worst case) theoretical Big-O analysis.

time complexity of WWL is $\tau(m) = \mathcal{O}(m^3 \log(m))$. The runtime of parallelized TMD is roughly three times larger than WWL on DD, which is much faster than the (worst case) theoretical Big-O analysis. In datasets contain small graphs such as NCI1, TMD without parallelization works well.

## 5 TMD and Stability of Graph Neural Networks

Next, we relate TMD to the perturbation stability of message passing GNNs. In particular, we use the Lipschitz constant, which relies on an underlying metric – here, a metric over graphs. We observe that TMD is a meaningful pseudometric in this case. The resulting Lipschitz allows to analyze the stability of GNNs under perturbations and generalization bounds under distribution shifts.

### 5.1 Lipschitz Constant of Message Passing Graph Neural Networks

For simplicity, we consider graph binary classification with the Graph Isomorphim Network (GIN) [55], one of the most widely applied and powerful GNNs. In particular, we consider the following message passing rules of a $L$-layer GIN:

$$\text{Message Passing} \quad z_v^{(l)} = \phi^{(l)}\left(z_v^{(l-1)} + \epsilon \sum_{u \in \mathcal{N}(v)} z_u^{(l-1)}\right), \quad \text{Graph Readout} \quad h(G) = \phi^{(L+1)}\left(\sum_{u \in V} z_u^{(L)}\right)$$

where $\phi^{(l)} : \mathbb{R}^d \to \mathbb{R}^d$ and $\phi^{(L+1)} : \mathbb{R}^d \to \mathbb{R}$ are learnable functions with Lipschitz constant $K_\phi^{(l)}$ and the initial state is set to the node feature $z_v^{(0)} = x_v$. The $\epsilon > 0$ is a weighting term between the center node and the neighbors. Note that the original formulation of GIN [55] weights the center node with layer dependent $\epsilon$ instead of neighbors. We adopt the form above with the purpose to simplify the notation of TMD. One can easily derive an equivalent form by changing the weight function $w(\cdot)$ of TMD to recover the original formulation of GIN. For simplicity, we set $\epsilon = 1$ in all experiments. This does not affect the empirically performance of GIN as the original paper shows [55]. A graph level binary classifier is constructed based on the logits $h(G)$.

The next theorem bounds the Lipschitz constant of GIN with respect to TMD. Although TMD is a pseudometric, it satisfies that $\text{TMD}_w^L(G_a, G_b) > 0$ if $G_a, G_b$ are distinguished by $L$ iterations of the WL test. Since $\text{GIN}(G_a) = \text{GIN}(G_b)$ for all graphs where WL fails, it holds that, if $\text{GIN}(G_a) \neq \text{GIN}(G_b)$ then $\text{TMD}_w^L(G_a, G_b) > 0$. That is, for all triples of graphs that GIN distinguishes pairwise, TMD is a metric.

**Theorem 8** (**Lipschitz Constant of GIN**). *Given an $L$-layer graph neural network $h : \mathcal{X} \to \mathbb{R}$ and two graph $G_a, G_b \in \mathcal{G}$, we have*

$$\left\| h(G_a) - h(G_b) \right\| \leq \prod_{l=1}^{L+1} K_\phi^{(l)} \cdot \text{TMD}_w^{L+1}(G_a, G_b),$$

*where $w(l) = \epsilon \cdot P_{L+1}^{l-1}/P_{L+1}^l$ for all $l \leq L$ and $P_L^l$ is the l-th number at level $L$ of Pascal's triangle.*

The result can be extended to other GNNs (Appendix B). Interestingly, the weights in each level follow a simple rule determined by Pascal's triangle. Figure 3 illustrates the weights when $L = 4$, where the weights $w(l)$ gradually decrease from $4\epsilon$ to $\epsilon/4$ as $l$ becomes smaller. Note that the hyperparameter $w$ of TMD is independent of the parameters of the GNN. Moreover, Theorem 8 shows that the Lipschitz constant of GNNs under TMD admits a simple form $\prod_{l=1}^{L+1} K_\phi^{(l)}$: the product of Lipschitz constants across layers, similar to fully connected networks [33, 37].

$$\begin{array}{ccccccccc}
 & & & & 1 & & & & \\
 & & & 1 & & 1 & & & \\
 & & 1 & & 2 & & 1 & & \\
 & 1 & & 3 & & 3 & & 1 & \\
1 & & 4 & & 6 & & 4 & & 1
\end{array}$$

$L = 4$ weights
$$\begin{array}{cccc}
\frac{1}{4} & \frac{4}{6} & \frac{6}{4} & \frac{4}{1}
\end{array}$$

Figure 3: **Example of Pascal's triangle** ($\epsilon = 1$).

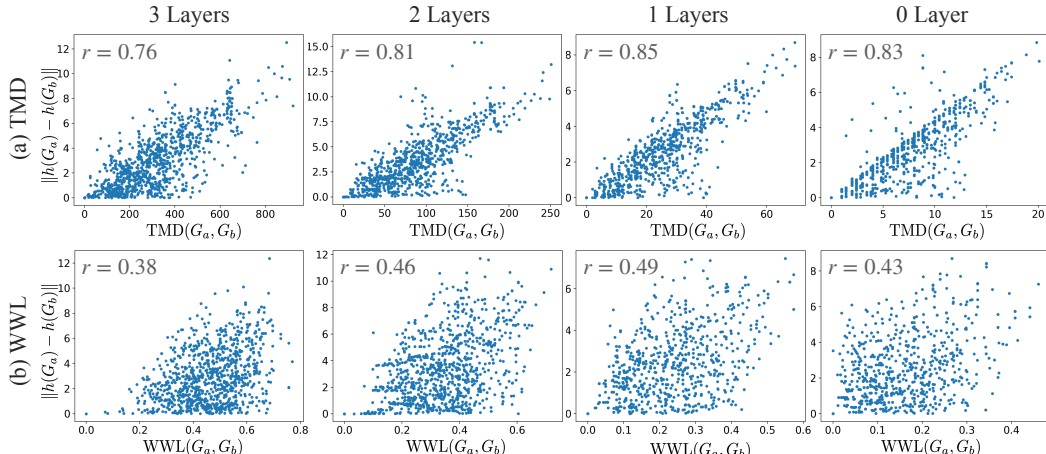

Figure 4: **Correlation between GNNs and TMD / WWL.** The Pearson correlation coefficient $r$ between $\|h(G_a) - h(G_b)\|$ and TMD / WWL are showed on the upper left of the figures. The output variation is highly correlated with TMD, while WWL barely captures the behavior of GNNs with different number of message-passing layers.

## 5.2 Stability of GNNs under Graph Perturbation

The Lipschitz constant is a common criterion to assess the stability of the neural networks to small perturbations [51]. Theorem 8 implies that the output variation of GNNs under graph perturbation can be bounded via the TMD between the original graph and the perturbed one. In this section, we analyze the stability of GNNs in more detail by dissecting TMD under three types of graph perturbation: (1) node drop; (2) edge drop; and (3) node feature perturbations.

**Proposition 9** (Node Drop). *Given a graph $G = (V, E)$, let $G'$ be the graph where node $v \in V$ is dropped. Then the tree mover's distance between $G$ and $G'$ can be bounded by*

$$\text{TMD}_w^L(G, G') \leq \sum_{l=1}^{L} \lambda_l \cdot \underbrace{\text{Width}_l(T_v^L)}_{\text{Tree Size}} \cdot \underbrace{\text{TD}_w(T_v^{L-l+1}, T_0)}_{\text{Tree Norm}},$$

*where* $\text{Width}_l(T)$ *is the width of $l$-th level of tree $T$ and* $\lambda_1 = 1$, $\lambda_l = \prod_{j=1}^{l-1} w(L + 1 - j)$.

The bound is controlled by two factors: (1) tree size and (2) tree norm (distance from the blank tree) of the computation tree $T_v^L$. A node $v$ with a large computation tree size implies that many nodes can be reached from $v$. Deleting $v$ from the graph then significantly changes the computation trees of those reachable nodes, while the magnitude of the variation is controlled by the tree norm.

**Proposition 10** (Edge Drop). *Given a graph $G = (V, E)$, let $G'$ be the graph where edge $(u, v) \in E$ is dropped. The tree mover's distance between $G$ and $G'$ can be bounded by*

$$\text{TMD}_w^L(G, G') \leq \sum_{l=1}^{L-1} \lambda_{l+1} \cdot \left( \text{Width}_l(T_v^L) \cdot \text{TD}_w(T_u^{L-l}, T_0) + \text{Width}_l(T_u^L) \cdot \text{TD}_w(T_v^{L-l}, T_0) \right).$$

The bound takes a similar form as with the node drop, but includes the effects of both node $v$ and $u$. In particular, the tree size of $v$ ($u$) will control how many computation trees of $u$ ($v$) will be dropped. By using Proposition 9 and 10, one can derive bounds for dropping multiple nodes and edges, or even *edge rewiring*. Note that adding nodes or edges is equivalent to the analysis above.

**Proposition 11** (Node Perturbation). *Given a graph $G = (V, E)$, let $G'$ be the graph where node feature $x_v$ is perturbed to $x'_v$. The tree mover's distance between $G$ and $G'$ is equal to*

$$\text{TMD}_w^L(G, G') \leq \sum_{l=1}^{L} \lambda_l \cdot \text{Width}_l(T_v^L) \cdot \|x_v - x'_v\|.$$

Different from Proposition 9, the magnitude of the perturbation is controlled by the norm of the perturbation instead of the norm of the computation tree.

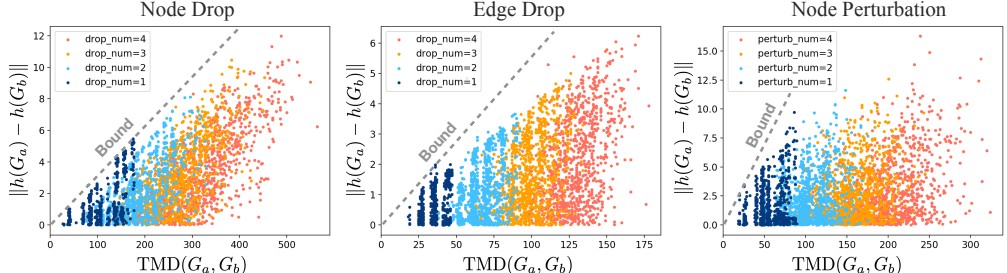

Figure 5: **Robustness under Graph Perturbation.** The empirical Lipschitz bounds (dash lines) successfully upper bound the output variation of GNNs under different graph perturbations.

### 5.3 Experiments

**Correlation between TMD and GNN output perturbation**   We now empirically examine the theoretical analysis with experiments on the MUTAG dataset [14]. Results for other datasets can be found in Appendix C. In particular, we measure to what extent candidate graph distances capture input perturbations that lead to output perturbations in the GNN. We train graph isomorphism networks [55] with varying numbers of message passing layers and plot the relation between input variation $\mathrm{TMD}_w^{L+1}(G_a, G_b)$ and output variation $\|h(G_a) - h(G_b)\|$ for randomly sampled pairs $(G_a, G_b)$ in Figure 4. For comparison, we also plot the input variations measured by the recently proposed graph metric WWL [46]. We can see that TMD strongly correlates with the output variation with large Pearson correlation coefficient, supporting the approach of defining the Lipschitz constant with respect to TMD, as in Theorem 8. In contrast, WWL barely captures the input graph perturbations that lead to output variation of GNNs.

**Stability under small Perturbations**   Next, we plot the output variation of 3-layer GNNs and the TMD under random graph perturbations in Figure 5. For node perturbations, we change the discrete node attribute for randomly sampled nodes. We additionally plot the Lipschitz bound with empirical estimated Lipschitz constant $\max_{G_a, G_b \in S} \|h(G_a, G_b)\| / \mathrm{TMD}_w^{L+1}(G_a, G_b)$, where $S$ is a set of samples. We refer reader to [36, 47] for analyses on approximation error of estimating Lipschitz constants from finite samples. We can see that the bound is reasonably tight and estimates the effect of perturbations across different degrees.

## 6   Generalization of GNNs under Distribution Shifts

Finally, we relate the Lipschitz condition of GNNs to the generalization error under distribution shifts by extending the results from [43]. Consider a binary classification task with input space $\mathcal{X}$ and output space $\mathcal{Y}$. In domain adaptation [13], a pair of source and target distributions $\mu_S, \mu_T$ over $\mathcal{X} \times \mathcal{Y}$ are given. Let $p_S$ and $p_T$ denote the respective marginals on the input space $\mathcal{X}$. In unsupervised domain adaptation, the learning algorithm obtains labelled source samples from $\mu_S$ and unlabelled target samples from $p_T$. To estimate the adaptability of a hypothesis $h$, i.e., its generalization to the target distribution, we aim to bound the target risk $R_T(h) = \mathbb{E}_{x,y \sim \mu_T}[\mathbb{1}_{h(x) \neq y}]$ relative to the source risk $R_S(h) = \mathbb{E}_{x,y \sim \mu_S}[\mathbb{1}_{h(x) \neq y}]$ [5, 13, 60]. For instance, Shen et al. [43] bound the target risk via the source risk and the Wasserstein-1 distance $\mathcal{W}_1$ between source and target distributions.

**Theorem 12** (Shen et al. [43]). *For all hypotheses $h \in \mathcal{H}$, the target risk is bounded as*

$$R_T(h) \leq R_S(h) + 2K\mathcal{W}_1(p_S, p_T) + \lambda_{\mathcal{H}},$$

*where $K$ is the Lipschitz constant of $h$ and $\lambda_{\mathcal{H}}$ is the best joint risk $\lambda_{\mathcal{H}} := \inf_{h' \in \mathcal{H}}[R_S(h') + R_T(h')]$.*

This bound relies on being able to measure Wasserstein distance between the two data distributions, which demands a ground metric on the data space, and an associated Lipschitz constant of the model $h$. The TMD and the resulting Lipschitz constant in Section 5.1 make this bound applicable to message passing GNNs, too. In particular, for GNNs that satisfy the Lipschitz constant in Theorem 8, the domain discrepancy $\mathcal{W}_1(p_S, p_T)$ is defined as

$$\mathcal{W}_1(p_S, p_T) = \inf_{\pi \in \Pi(p_S, p_T)} \int \mathrm{TMD}_w^{L+1}(G_a, G_b) d\pi(G_a, G_b), \tag{3}$$

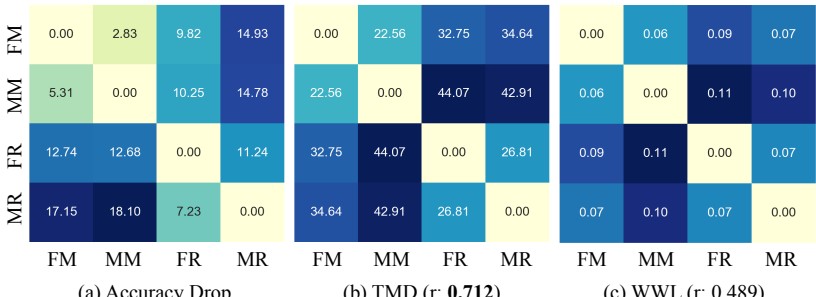

| | FM | MM | FR | MR | | FM | MM | FR | MR | | FM | MM | FR | MR |
|---|---|---|---|---|---|---|---|---|---|---|---|---|---|---|
| **FM** | 0.00 | 2.83 | 9.82 | 14.93 | | 0.00 | 22.56 | 32.75 | 34.64 | | 0.00 | 0.06 | 0.09 | 0.07 |
| **MM** | 5.31 | 0.00 | 10.25 | 14.78 | | 22.56 | 0.00 | 44.07 | 42.91 | | 0.06 | 0.00 | 0.11 | 0.10 |
| **FR** | 12.74 | 12.68 | 0.00 | 11.24 | | 32.75 | 44.07 | 0.00 | 26.81 | | 0.09 | 0.11 | 0.00 | 0.07 |
| **MR** | 17.15 | 18.10 | 7.23 | 0.00 | | 34.64 | 42.91 | 26.81 | 0.00 | | 0.07 | 0.10 | 0.07 | 0.00 |
| | (a) Accuracy Drop | | | | | (b) TMD (r: **0.712**) | | | | | (c) WWL (r: 0.489) | | | |

Figure 6: **Accuracy Drop and Distances.** Wasserstein distance based on TMD highly correlates ($r = 0.712$) with the accuracy drops, while WWL fails to predict the generalization.

where $L$ is the number of message-passing layers in $h$. Since $\mathcal{W}_1(p_S, p_T)$ can be estimated without labels, Theorem 12 applies to unsupervised domain adaptation. Assuming that there is a model that performs well in both source and target domain, i.e., $\lambda_{\mathcal{H}}$ is small, we may empirically estimate the discrepancy between source and target risk via the GNN Lipschitz constant and $\mathcal{W}_1(p_S, p_T)$.

### 6.1 Experiments

**Domain Shifts** We first verify our analysis on the PTC dataset [22], which contains carcinogenicity labels of chemical structures for four groups of rodents: male mice (MM), male rats (MR), female mice (FM) and female rats (FR). We train 3-layer GINs [55] on one group and examine the empirical performance drop on the remaining groups. Figure 6 shows the Wasserstein distance between groups and the corresponding performance drops. As a baseline, we compute the Wasserstein distance with WWL transportation cost [46]. The $\mathcal{W}_1$ distance based on TMD highly correlates with the accuracy drop (Pearson correlation $r = 0.712$), while WWL-$\mathcal{W}_1$ only achieves $r = 0.489$.

**Size Generalization** A known challenge for GNNs is generalizing to graphs of different size [58, 56]. To evaluate TMD on this problem, we sort the PROTEINS dataset [8] based on the number of nodes and bin it into 8 subsets, each containing 125 graphs. We again train 3-layer GINs on the smallest and the largest bins and examine the accuracy drops on the remaining ones. Figure 7 plots the accuracy drops with respect to different subsets and the corresponding Wasserstein distance based on TMD and WWL. TMD correlates with the accuracy drop surprisingly well when the models are trained on large graphs and tested on smaller graphs ($r = \mathbf{0.97}$). The correlation is slightly weaker when the models are trained on small graphs and tested on large graphs ($r = 0.83$). Yet, TMD shows a much more gradual change in distance, in agreement with the gradual change in drop, than WWL, and hence reflects the overall behavior better.

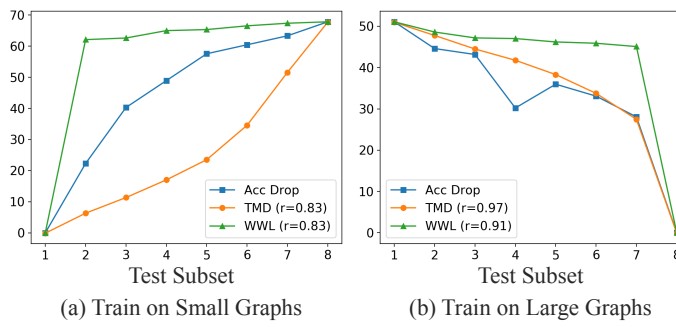

(a) Train on Small Graphs — (b) Train on Large Graphs

Figure 7: **Size Generalization.** Index 1 denotes the bin of smallest graphs and index 8 the largest graphs. For better visualization, the Wasserstein distances are normalized to make the maximal distance equal to the greatest performance drops, as the absolute scales of distances are less important. The models in (a) / (b) are trained on subset 1 / 8, respectively.

## 7 Conclusion

In this work, we introduce Tree Mover's Distance (TMD), a new graph distance based on optimal transport between computation trees. First, TMD captures the structural and attribute properties that are important for many graph classification tasks. Second, it reflects the patterns that determine the generalization behavior of message passing graph neural networks, and, hence, offers a suitable tool to predict the perturbation stability and out-of-domain generalization capability of such GNNs. Hence, it bears promise in applications, both for graph learning tasks and predicting reliability of graph learning models, and, in theory, as a tool for new tighter analyses of generalization in GNNs.

**Acknowledgements**   This work was in part supported by NSF BIGDATA IIS-1741341, NSF AI Institute TILOS, NSF CAREER 1553284. CC is supported by a IBM PhD Fellowship.

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
