## Broader Impact

Graph Neural Networks are used in many applications with potential societal implications: predictions on social networks, drug design, computational chemistry and materials science, traffic predictions, etc. In many of these applications, the model may be faced with distribution shifts that may lead to a decline in performance. This work is a step towards understanding and estimating such behavior, by understanding and formalizing what kinds of distribution shifts may impact the model.

Indeed, robustness is closely related to fairness, when the distribution shifts are associated with different demographic groups [12, 4, 59]. Among the above applications, this may be particularly an issue with social network analysis, and possibly drug design and traffic prediction. In that case, the results in this paper may provide a basis for diagnostic tools be enabling to quantify the amount of potentially risky distribution shifts.

## A   Proof

### A.1   Preliminaries

**Equivalence between OT and Wasserstein Distance**   Throughout the proof, we will frequently use the equivalence between optimal transport and Wasserstein distance to simplify the notation. Following the setting in section 3, let $X = \{x_i\}_{i=1}^m$ and $Y = \{y_i\}_{j=1}^m$ be two multisets, both containing $m$ elements, and let $\mathcal{U}(X)$ denote the uniform distribution over a multiset $X$. We first note the following equivalence:

$$\mathrm{OT}_d(X,Y) := \min_{\gamma \in \Gamma(X,Y)} \langle C, \gamma \rangle = m \cdot \mathrm{OT}_d^*(X,Y) = m \cdot \mathcal{W}_d(\mathcal{U}(X),\mathcal{U}(Y))),$$

where $\mathcal{W}_d(P,Q)$ is the Wasserstein distance between $P,Q$ with cost function $d$ defined as follows:

$$\mathcal{W}_d(P,Q) = \inf_{\pi \in \Pi(P,Q)} \int d(x,y) d\pi(x,y).$$

The $\Pi$ denotes the set of measure couplings whose marginals are $P$ and $Q$, respectively.

**Invariance to Additional Augmentation**   Here we show that unnormalized OT is invariant to "blank" augmentation, which is an important property for proving the main theorem.

**Lemma 13.** *Assume we are given two multisets $X = \{x_i\}_{i=1}^m, Y = \{y_j\}_{j=1}^m$ with the same cardinality $m$, where $x_i, y_j \in \mathcal{X}$ for all $i, j$. Let $d$ be a metric on $\mathcal{X}$ and $\mathbb{0} \in \mathcal{X}$. Then we have*

$$\mathrm{OT}_d(X \cup (\mathbb{0})^n, Y \cup (\mathbb{0})^n) = \mathrm{OT}_d(X,Y).$$

*Proof.* First, by construction, we already have

$$\mathrm{OT}_d(X \cup (\mathbb{0})^n, Y \cup (\mathbb{0})^n) \leq \mathrm{OT}_d(X,Y),$$

as we can always keep the original coupling padded with an identity matrix and get the same cost. Now suppose we adopt a new coupling for $\mathrm{OT}_d(X \cup (\mathbb{0})^n, Y \cup (\mathbb{0})^n)$ which might lead to smaller cost. Note that any change to the coupling can be decomposed into two steps: (1) permute, change the permutation in the original coupling and (2) decouple, change the coupling $(x - y)$ to $(x - \mathbb{0}), (y - \mathbb{0})$ for some pairing $(x, y)$. In the permutation step, the cost is always increasing as we are destroy the original optimal coupling. In the second step, the cost will also increase due to the triangle inequality of cost function $d$:

$$d(x,y) + d(\mathbb{0},\mathbb{0}) = d(x,y) \leq d(x,\mathbb{0}) + d(y,\mathbb{0}).$$

Therefore, we prove that the equality case must hold by contradiction. Note that the proof holds for any augmentation, not only zero vectors. $\qquad\square$

**Unnormalized OT is a Metric for Multisets**   Finally, we show that if the transportation cost $d$ is a pseudometric, then the unnormalized OT with blank augmentation $\rho$ is also a pseudometric for multisets.

**Lemma 14.** *If $d$ is a metric, then $\mathrm{OT}_d(\rho(\cdot,\cdot))$ is a metric over multisets that do not contain $\mathbb{0}$. If $\mathbb{0}$ is included, $\mathrm{OT}_d(\cdot,\cdot)$ is a pseudometric.*

*Proof.* The first two axioms, $\mathrm{OT}_d(\rho(X,X)) = 0$ if and only if $X = X$ and $\mathrm{OT}_d(\rho(X,Y)) = \mathrm{OT}_d(\rho(Y,X))$ immediately hold via the property of optimal transport and Wasserstein distance if $\mathbb{0} \notin X, Y$. The constraint on $\mathbb{0}$ is due to the fact $\mathrm{OT}_d(\rho(X, X \cup \mathbb{0})) = 0$, which violates the first axiom. But practically the augmentations are specified to be different from the elements in the sets.

Next, we prove the Triangle inequality of unnormalized OT. In particular, we will leverage the equivalence of unnormalized OT and Wasserstein distance as follows:

$$\mathrm{OT}_d(\rho(X,Y))$$

$$= \max(|X|,|Y|) \cdot \mathcal{W}_d\left(\mathcal{U}\left(X\bigcup(\mathbb{0})^{\max(|Y|-|X|,0)}\right), \mathcal{U}\left(Y\bigcup(\mathbb{0})^{\max(|X|-|Y|,0)}\right)\right)$$

$$= \max(|X|,|Y|,|Z|) \cdot \mathcal{W}_d\left(\mathcal{U}\left(X\bigcup(\mathbb{0})^{\max(\max(|Y|,|Z|)-|X|,0)}\right), \mathcal{U}\left(Y\bigcup(\mathbb{0})^{\max(\max(|X|,|Z|)-|Y|,0)}\right)\right)$$
$$\text{(Lemma 13)}$$

$$\leq \max(|X|,|Y|,|Z|) \cdot \left(\mathcal{W}_d\left(\mathcal{U}\left(X\bigcup(\mathbb{0})^{\max(\max(|Y|,|Z|)-|X|,0)}\right), \mathcal{U}\left(Z\bigcup(\mathbb{0})^{\max(\max(|X|,|Y|)-|Z|,0)}\right)\right)\right.$$

$$\left. + \mathcal{W}_d\left(\mathcal{U}\left(Z\bigcup(\mathbb{0})^{\max(\max(|X|,|Y|)-|Z|,0)}\right), \mathcal{U}\left(Y\bigcup(\mathbb{0})^{\max(\max(|X|,|Z|)-|Y|,0)}\right)\right)\right)$$
$$\text{(Triangle Ineq of } \mathcal{W})$$

$$= \max(|X|,|Z|) \cdot \mathcal{W}_d\left(\mathcal{U}\left(X\bigcup(\mathbb{0})^{\max(|Z|-|X|,0)}\right), \mathcal{U}\left(Z\bigcup(\mathbb{0})^{\max(|X|-|Z|,0)}\right)\right)$$

$$+ \max(|Z|,|Y|) \cdot \mathcal{W}_d\left(\mathcal{U}\left(Z\bigcup(\mathbb{0})^{\max(|Y|-|Z|,0)}\right), \mathcal{U}\left(Y\bigcup(\mathbb{0})^{\max(|Z|-|Y|,0)}\right)\right)$$
$$\text{(Lemma 13)}$$

$$= \mathrm{OT}_d(\rho(X,Z)) + \mathrm{OT}_d(\rho(Z,Y))$$

The inequality holds as Wasserstein distance is a pseudometric if the transportation cost is a pseudometric [45, 3]. □

### A.2 Tree Mover's Distance is a Pseudometric

Via Lemma 14, to show TMD is a pseudometric, we will focus on proving that the transportation cost TD is a pseudometric between two trees.

**Lemma 15.** *The distance $\mathrm{TD}_w(\rho(\cdot,\cdot))$ is a metric for two rooted trees if $w(\cdot) > 0$ and node features do not contain zero vectors, where $\rho$ is the blank tree augmentation. Otherwise, it is a pseudometric.*

*Proof.* We first prove that $\mathrm{TD}_w(T_a, T_b) = 0$ if and only if $T_a = T_b$ by induction. In particular, we will focus on the case $\mathrm{Depth}(T_a) = \mathrm{Depth}(T_b)$, otherwise the statement trivially holds. When depth $= 1$, the $\mathrm{TD}_w$ reduces to Euclidean distance, which is a metric. Suppose the statement holds for depth-$k$ trees. Given two tree $T_a, T_b$ with depth equals to $k + 1$, their distance is

$$\mathrm{TD}_w(T_a, T_b) = \|r_{T_a} - r_{T_b}\| + w(\mathrm{Depth}(T_a)) \cdot \mathrm{OT}_{\mathrm{TD}_w}\left(\rho(\mathcal{T}_{r_{T_a}}, \mathcal{T}_{r_{T_b}})\right).$$

Since $\mathrm{TD}_w$ is a metric for depth-k tree by assumption, and $\mathrm{OT}_{\mathrm{TD}_w}(\rho(\cdot,\cdot))$ is a metric if $\mathrm{TD}_w$ is a metric via Lemma 14, $\mathrm{TD}_w(T_a, T_b) = 0$ if and only if $r_{T_a} = r_{T_b}$ and $\mathcal{T}_{r_{T_a}} = \mathcal{T}_{r_{T_b}}$, which completes the proof of induction. Nevertheless, similar to Lemma 14, this does not hold if the nodes contains $\mathbb{0}$, which should not be the case practically. Otherwise, one can simply add a small values to node features to distinguish them from the zero vector.

For the triangle inequality, we give a proof by induction on depth. For the base case depth $= 1$, the tree mover's distance is simply the Wasserstein-1 distance between the distribution of augmented node feature vectors and the transportation cost $\mathrm{TD}_w(\cdot,\cdot)$ reduces to the Euclidean distance, which is a metric and satisfies the Triangle inequality. Next, we show that if $\mathrm{TD}_w(\cdot,\cdot)$ satisfies the Triangle

ineuqality for depth-k trees, it is also a metric for depth-(k+1) trees. For trees $T_a$ and $T_b$ with depth $k + 1$, introduced the third tree $T_c$ with depth $k + 1$, we have

$$
\begin{aligned}
\text{TD}_w(T_a, T_b) =& \|r_{T_a} - r_{T_b}\| + w(k+1)\text{OT}_{\text{TD}_w}\left(\rho(\mathcal{T}_{r_{T_a}}, \mathcal{T}_{r_{T_b}})\right) \\
\leq& \|r_{T_a} - r_{T_c}\| + \|r_{T_c} - r_{T_b}\| + w(k+1)\text{OT}_{\text{TD}_w}\left(\rho(\mathcal{T}_{r_{T_a}}, \mathcal{T}_{r_{T_b}})\right) \\
\leq& \|r_{T_a} - r_{T_c}\| + \|r_{T_c} - r_{T_b}\| + w(k+1)\text{OT}_{\text{TD}_w}\left(\rho(\mathcal{T}_{r_{T_a}}, \mathcal{T}_{r_{T_c}})\right) \\
& + w(k+1)\text{OT}_{\text{TD}_w}\left(\rho(\mathcal{T}_{r_{T_c}}, \mathcal{T}_{r_{T_b}})\right) \qquad \text{(Induction hypothesis)} \\
=& \text{TD}_w(T_a, T_c) + \text{TD}_w(T_c, T_b).
\end{aligned}
$$

The second inequality holds since OT is a pseudometric for multisets over depth-$k$ trees as $\text{TD}_w(\cdot, \cdot)$ satisfies the Triangle ineuqality for depth-$k$ trees via induction hypothesis. Since $\text{TD}_w$ satisfies the triangle inequality for depth-$k + 1$ trees, via Lemma 14, TMD also satisfies the triangle inequality, which completes the proof by mathematical induction. $\qquad\square$

## A.3 WL Proof

**Theorem 16** (Discriminative Power of TMD). *If two graphs $G_a = (V_a, E_a)$, $G_b = (V_b, E_b)$ are determined to be non-isomorphic in WL iteration $L$ and $w(l) > 0$ for all $0 < l \leq L + 1$ and $\mathbb{0} \notin V_a, V_b$, then $\text{TMD}_w^{L+1}(G_a, G_b) > 0$.*

*Proof.* We will show that if two nodes have the same subtree, then their WL labels will be the same by induction. The statement holds when depth is 1, as all the WL labels are the same and all the subtrees have the same single node. Suppose the statement holds for depth-$k$ tree. If WL identifies two graphs are non-isomorphic at iteration $k + 1$, this means there are at least a pair of nodes whose neighbors have different WL label at $k$-th iteration. This implies that the neightbors have different subtree via induction hypothesis. Therefore, the depth-$k + 1$ subtree of the node, which is constructed by appending the subtrees of neightbors to a new node, will also be different. Therefore, the TMD between the subtrees will be greater than zero via Lemma 15, implying that $\text{TMD}_\lambda^{k+1}$ also determine two graphs are non-isomorphic for all $\lambda > 1$. Again, we add the zero vector constraint for the same reason as Lemma 14 and 15. $\qquad\square$

## A.4 Lipschitz Bound of GNN proof

*Proof.* For simplicity, we set $\epsilon = 1$ through out the proof. We first bound the difference in prediction based on the embedding in the last layer:

$$
\begin{aligned}
\left\|h(G_a) - h(G_b)\right\| &= \left\|\phi^{(L+1)}\left(\sum_{i \in V_a} z_i^{(L)}\right) - \phi^{(L+1)}\left(\sum_{j \in V_b} z_j^{(L)}\right)\right\| \\
&\leq K_\phi^{(L+1)} \cdot \left\|\sum_{i \in V_a} z_i^{(L)} - \sum_{j \in V_b} z_j^{(L)}\right\|. \qquad \text{(Definition of Lipschitz Condition)}
\end{aligned}
$$

Let $V_a^\rho$ and $V_b^\rho$ be two multisets of nodes after blank tree augmentation: $(V_a^\rho, V_b^\rho) = \rho(V_a, V_b)$, we have

$$
\left\|\sum_{i \in V_a} z_i^{(L)} - \sum_{j \in V_b} z_j^{(L)}\right\| = \left\|\sum_{i \in V_a^\rho, j \in V_b^\rho} T_{i,j}^{(L, V_a V_b)}\left(z_i^{(L)} - z_j^{(L)}\right)\right\| \tag{4}
$$

$$
\leq \sum_{i \in V_a^\rho, j \in V_b^\rho} T_{i,j}^{(L, V_a V_b)}\left\|z_i^{(L)} - z_j^{(L)}\right\| \tag{5}
$$

where $T^{(L,V_a^\rho V_b^\rho)}$ is a transportation plan between $V_a^\rho$ and $V_b^\rho$. Note that the inequality holds for any valid transportation plan. Let $\Delta^{(L)}_{T^{(L,V_a^\rho V_b^\rho)}} = \sum_{i \in V_a^\rho, j \in V_b^\rho} T_{i,j}^{(L,V_a^\rho V_b^\rho)} \left\| z_i^{(L)} - z_j^{(L)} \right\|$, we have

$$
\begin{aligned}
&\Delta^{(L)}_{T^{(L,V_a^\rho V_b^\rho)}} \\
&\leq \sum_{i \in V_a^\rho, j \in V_b^\rho} T_{i,j}^{(L,V_a^\rho V_b^\rho)} \left[ \left\| \phi^{(L)}\left( z_i^{(L-1)} + \sum_{i' \in \mathcal{N}(i)} [z_i^{(L-1)}] \right) - \phi^{(L)}\left( z_j^{(L-1)} + \sum_{j' \in \mathcal{N}(i)} [z_j^{(L-1)}] \right) \right\| \right] \\
&\leq K_\phi^{(l)}\left( \sum_{i \in V_a^\rho, j \in V_b^\rho} T_{i,j}^{(L,V_a^\rho V_b^\rho)} \left[ \left\| z_i^{(L-1)} - z_j^{(L-1)} \right\| + \left\| \sum_{i' \in \mathcal{N}(i)} z_i^{(L-1)} - \sum_{j' \in \mathcal{N}(i)} z_j^{(L-1)} \right\| \right] \right) \\
&\leq K_\phi^{(l)}\left( \sum_{i \in V_a^\rho, j \in V_b^\rho} T_{i,j}^{(L,V_a^\rho V_b^\rho)} \left[ \left\| z_i^{(L-1)} - z_j^{(L-1)} \right\| + \sum_{i' \in \mathcal{N}(i)^\rho, j' \in \mathcal{N}(j)^\rho} T_{i',j'}^{(L-1,\mathcal{N}(i)^\rho \mathcal{N}(j)^\rho)} \left[ \left\| z_{i'}^{(L-1)} - z_{j'}^{(L-1)} \right\| \right] \right] \right) \\
&= K_\phi^{(l)}\left( \sum_{i \in V_a^\rho, j \in V_b^\rho} T_{i,j}^{(L,V_a^\rho V_b^\rho)} \left[ \left\| z_i^{(L-1)} - z_j^{(L-1)} \right\| + \Delta^{(L-1)}_{T^{(L-1,\mathcal{N}(i)^\rho \mathcal{N}(j)^\rho)}} \right] \right)
\end{aligned}
$$

Here we introduce another transportation plan $T_{i',j'}^{(L-1,\mathcal{N}(i)^\rho \mathcal{N}(j)^\rho)}$ to pair augmented $\mathcal{N}(i)^\rho$ and $\mathcal{N}(j)^\rho$ for all $i \in V_a^\rho, j \in V_b^\rho$ at layer $L-1$. We then further bound first term using similar strategy:

$$
\begin{aligned}
&\sum_{i \in V_a^\rho, j \in V_b^\rho} T_{i,j}^{(L,V_a V_b)} \left[ \left\| z_i^{(L-1)} - z_j^{(L-1)} \right\| \right] \\
&= \sum_{i \in V_a^\rho, j \in V_b^\rho} T_{i,j}^{(L,V_a V_b)} \left[ \left\| \phi^{(L-1)}\left( z_i^{(L-2)} + \sum_{i' \in \mathcal{N}(i)} z_{i'}^{(L-2)} \right) - \phi^{(L-1)}\left( z_j^{(L-2)} + \sum_{j' \in \mathcal{N}(j)} z_{j'}^{(L-2)} \right) \right\| \right] \\
&\leq K_\phi^{(L-1)}\left( \sum_{i \in V_a^\rho, j \in V_b^\rho} T_{i,j}^{(L,V_a V_b)} \left( \left\| z_i^{(L-2)} - z_j^{(L-2)} \right\| + \sum_{i' \in \mathcal{N}(i)^\rho, j' \in \mathcal{N}(j)^\rho} T_{i',j'}^{(L-1,\mathcal{N}(i)^\rho \mathcal{N}(j)^\rho)} \left\| z_{i'}^{(L-2)} - z_{j'}^{(L-2)} \right\| \right) \right). \\
&= K_\phi^{(L-1)}\left( \sum_{i \in V_a^\rho, j \in V_b^\rho} T_{i,j}^{(L,V_a V_b)} \left( \left\| z_i^{(L-2)} - z_j^{(L-2)} \right\| + \Delta^{(L-2)}_{T^{(L-1,\mathcal{N}(i)^\rho \mathcal{N}(j)^\rho)}} \right) \right).
\end{aligned}
$$

Note that we still use the same transportation plan $T_{i',j'}^{(L-1,\mathcal{N}(i)^\rho \mathcal{N}(j)^\rho)}$ to bound the sum difference. Apply this step recursively, the first term will eventually become $\|x_i - x_j\|$ as

$$
K_\phi^{(L)} \left\| z_i^{(L-1)} - z_j^{(L-1)} \right\| \to K_\phi^{(L)} K_\phi^{(L-1)} \left\| z_i^{(L-2)} - z_j^{(L-2)} \right\| \to \prod_{m=1}^{L} K_\phi^{(m)} \left\| z_i^{(0)} - z_j^{(0)} \right\| = \prod_{m=1}^{L} K_\phi^{(m)} \|x_i - x_j\|,
$$

and the bound will become

$$\Delta^{(l)}_{\pi^l_{V_a,V_b}} \leq \sum_{i \in V_a^\rho, j \in V_b^\rho} T^{(L,V_a V_b)}_{i,j} \left[ \left( \prod_{m=1}^L K^{(m)}_\phi \right) \|x_i - x_j\| + \sum_{k=1}^L \left( \prod_{m=1}^k K^{(L+1-m)}_\phi \right) \cdot \Delta^{(L-k)}_{T^{(L-1,\mathcal{N}(i)^\rho \mathcal{N}(j)^\rho)}} \right]$$

$$= \mathbb{E}_{(i,j) \sim \pi^l_{V_a,V_b}} \left[ \left( \prod_{m=1}^L K^{(m)}_\phi \right) \|x^i - x^j\| \right.$$

$$+ \underbrace{K^{(L)}_\phi \sum_{i' \in \mathcal{N}(i)^\rho, j' \in \mathcal{N}(j)^\rho} T^{(L-1,\mathcal{N}(i)^\rho \mathcal{N}(j)^\rho)}_{i',j'} \left\| z^{(L-1)}_{i'} - z^{(L-1)}_{j'} \right\|}_{①}$$

$$+ \underbrace{K^{(L)}_\phi K^{(L-1)}_\phi \sum_{i' \in \mathcal{N}(i)^\rho, j' \in \mathcal{N}(j)^\rho} T^{(L-1,\mathcal{N}(i)^\rho \mathcal{N}(j)^\rho)}_{i',j'} \| z^{(L-2)}_{i'} - z^{(L-2)}_{j'} \|}_{②}$$

$$\cdots$$

$$\left. + \underbrace{K^{(L)}_\phi K^{(L-1)}_\phi \cdots K^{(1)}_\phi \sum_{i' \in \mathcal{N}(i)^\rho, j' \in \mathcal{N}(j)^\rho} T^{(L-1,\mathcal{N}(i)^\rho \mathcal{N}(j)^\rho)}_{i',j'} \|x_{i'} - x_{j'}\|}_{Ⓛ} \right].$$

Next, we will bound ①, ②, $\cdots$ to ⓛ using a similar way as described above. To start, we introduce the second coupling $\pi^{l-2}_{\mathcal{N}(i'),\mathcal{N}(j')}$ for all $i', j' \sim \pi^{l-1}_{\mathcal{N}(i),\mathcal{N}(j)}$:

$$① \leq K^{(L)}_\phi \sum_{i' \in \mathcal{N}(i)^\rho, j' \in \mathcal{N}(j)^\rho} T^{(L-1,\mathcal{N}(i)^\rho \mathcal{N}(j)^\rho)}_{i',j'} \left[ K^{(L-1)}_\phi K^{(L-2)}_\phi \cdots K^{(1)}_\phi \|x_{i'} - x_{j'}\| \right.$$

$$+ K^{(L-1)}_\phi \sum_{i'' \in \mathcal{N}(i')^\rho, j'' \in \mathcal{N}(j')^\rho} T^{(L-2,\mathcal{N}(i')^\rho \mathcal{N}(j')^\rho)}_{i'',j''} \left[ \|z^{L-2}_{i''} - z^{L-2}_{j''}\| \right]$$

$$+ K^{(L-1)}_\phi K^{(L-2)}_\phi \sum_{i'' \in \mathcal{N}(i')^\rho, j'' \in \mathcal{N}(j')^\rho} T^{(L-2,\mathcal{N}(i')^\rho \mathcal{N}(j')^\rho)}_{i'',j''} \left[ \|z^{L-3}_{i''} - z^{L-3}_{j''}\| \right]$$

$$\cdots$$

$$\left. + K^{(L-1)}_\phi K^{(L-2)}_\phi \cdots K^{(1)}_\phi \sum_{i'' \in \mathcal{N}(i')^\rho, j'' \in \mathcal{N}(j')^\rho} T^{(L-2,\mathcal{N}(i')^\rho \mathcal{N}(j')^\rho)}_{i'',j''} \left[ \|x_{i''} - x_{j''}\| \right] \right]$$

$$:= bound(①)$$

$$\textcircled{2} \leq K_\phi^{(L)} K_\phi^{(L-1)} \sum_{i' \in \mathcal{N}(i)^\rho, j' \in \mathcal{N}(j)^\rho} T_{i',j'}^{(L-1,\mathcal{N}(i)^\rho \mathcal{N}(j)^\rho)} \Bigg[ K_\phi^{(L-2)} K_\phi^{(L-1)} \cdots K_\phi^{(1)} \|x_{i'} - x_{j'}\|$$

$$+ K_\phi^{(L-2)} \sum_{i'' \in \mathcal{N}(i')^\rho, j'' \in \mathcal{N}(j')^\rho} T_{i'',j''}^{(L-2,\mathcal{N}(i')^\rho \mathcal{N}(j')^\rho)} \left[ \|z_{i''}^{L-3} - z_{j''}^{L-3}\| \right]$$

$$+ K_\phi^{(L-2)} K_\phi^{(L-3)} \sum_{i'' \in \mathcal{N}(i')^\rho, j'' \in \mathcal{N}(j')^\rho} T_{i'',j''}^{(L-2,\mathcal{N}(i')^\rho \mathcal{N}(j')^\rho)} \left[ \|z_{i''}^{L-4} - z_{j''}^{L-4}\| \right]$$

$$\cdots$$

$$+ K_\phi^{(L-2)} K_\phi^{(L-3)} \cdots K_\phi^{(1)} \sum_{i'' \in \mathcal{N}(i')^\rho, j'' \in \mathcal{N}(j')^\rho} T_{i'',j''}^{(L-2,\mathcal{N}(i')^\rho \mathcal{N}(j')^\rho)} \left[ \|x_{i''} - x_{j''}\| \right] \Bigg]$$

$$= K_\phi^{(L)} \sum_{i'' \in \mathcal{N}(i')^\rho, j'' \in \mathcal{N}(j')^\rho} T_{i'',j''}^{(L-2,\mathcal{N}(i')^\rho \mathcal{N}(j')^\rho)} \Bigg[ K_\phi^{(L-1)} K_\phi^{(L-2)} \cdots K_\phi^{(1)} \|x_{i'} - x_{j'}\|$$

$$+ K_\phi^{(L-1)} K_\phi^{(L-2)} \sum_{i'' \in \mathcal{N}(i')^\rho, j'' \in \mathcal{N}(j')^\rho} T_{i'',j''}^{(L-2,\mathcal{N}(i')^\rho \mathcal{N}(j')^\rho)} \left[ \|z_{i''}^{L-3} - z_{j''}^{L-3}\| \right]$$

$$+ K_\phi^{(L-1)} K_\phi^{(L-2)} K_\phi^{(L-3)} \sum_{i'' \in \mathcal{N}(i')^\rho, j'' \in \mathcal{N}(j')^\rho} T_{i'',j''}^{(L-2,\mathcal{N}(i')^\rho \mathcal{N}(j')^\rho)} \left[ \|z_{i''}^{L-4} - z_{j''}^{L-4}\| \right]$$

$$\cdots$$

$$+ K_\phi^{(L-1)} K_\phi^{(L-2)} \cdots K_\phi^{(1)} \sum_{i'' \in \mathcal{N}(i')^\rho, j'' \in \mathcal{N}(j')^\rho} T_{i'',j''}^{(L-2,\mathcal{N}(i')^\rho \mathcal{N}(j')^\rho)} \left[ \|x_{i''} - x_{j''}\| \right] \Bigg]$$

$$:= bound(\textcircled{2})$$

We can see that the bounds for $\textcircled{1}$ and $\textcircled{2}$ only differ in the term $K_\phi^{(l-1)} \sum_{i'' \in \mathcal{N}(i')^\rho, j'' \in \mathcal{N}(j')^\rho} T_{i'',j''}^{(L-2,\mathcal{N}(i')^\rho \mathcal{N}(j')^\rho)} \left[ \|z_{i''}^{L-2} - z_{j''}^{L-2}\| \right]$. Moreover, one can show that $bound(\textcircled{k})$ and $bound(\textcircled{k+1})$ only differ in $K_\phi^{(l-1)} \cdots K_\phi^{(l-k)} \sum_{i'' \in \mathcal{N}(i')^\rho, j'' \in \mathcal{N}(j')^\rho} T_{i'',j''}^{(L-2,\mathcal{N}(i')^\rho \mathcal{N}(j')^\rho)} \left[ \|z_{i''}^{L-k-1} - z_{j''}^{L-k-1}\| \right]$. Therefore, we can merge bound $\textcircled{1}$ to bound $\textcircled{l}$ and get

$$\sum_{i=1}^{l} bound\,\textcircled{i} = K_\phi^{(L)} \sum_{i' \in \mathcal{N}(i)^\rho, j' \in \mathcal{N}(j)^\rho} T_{i',j'}^{(L-1,\mathcal{N}(i)^\rho \mathcal{N}(j)^\rho)} \Bigg[ L \cdot K_\phi^{(L-1)} K_\phi^{(L-2)} \cdots K_\phi^{(1)} \|x_{i'} - x_{j'}\|$$

$$+ K_\phi^{(L-1)} \sum_{i'' \in \mathcal{N}(i')^\rho, j'' \in \mathcal{N}(j')^\rho} T_{i'',j''}^{(L-2,\mathcal{N}(i')^\rho \mathcal{N}(j')^\rho)} \left[ \|z_{i''}^{L-2} - z_{j''}^{L-2}\| \right]$$

$$+ 2K_\phi^{(L-1)} K_\phi^{(L-2)} \sum_{i'' \in \mathcal{N}(i')^\rho, j'' \in \mathcal{N}(j')^\rho} T_{i'',j''}^{(L-2,\mathcal{N}(i')^\rho \mathcal{N}(j')^\rho)} \left[ \|z_{i''}^{L-3} - z_{j''}^{L-3}\| \right]$$

$$\cdots$$

$$+ (L-1) K_\phi^{(L-1)} K_\phi^{(L-2)} \cdots K_\phi^{(1)} \sum_{i'' \in \mathcal{N}(i')^\rho, j'' \in \mathcal{N}(j')^\rho} T_{i'',j''}^{(L-2,\mathcal{N}(i')^\rho \mathcal{N}(j')^\rho)} \left[ \|x_{i''} - x_{j''}\| \right] \Bigg]$$

Plugging the bound back yields

$$
\begin{aligned}
= \sum_{i \in V_a^\rho, j \in V_b^\rho} T_{i,j}^{(L,V_a V_b)} \Bigg[ & \left( \prod_{m=1}^{L} K_\phi^{(m)} \right) \|x^i - x^j\| \\
& + K_\phi^{(L)} \sum_{i' \in \mathcal{N}(i)^\rho, j' \in \mathcal{N}(j)^\rho} T_{i',j'}^{(L-1,\mathcal{N}(i)^\rho \mathcal{N}(j)^\rho)} \Big[ L \cdot K_\phi^{(L-1)} K_\phi^{(L-2)} \cdots K_\phi^{(1)} \|x_{i'} - x_{j'}\| \\
& \underbrace{+ K_\phi^{(L-1)} \sum_{i'' \in \mathcal{N}(i')^\rho, j'' \in \mathcal{N}(j')^\rho} T_{i'',j''}^{(L-2,\mathcal{N}(i')^\rho \mathcal{N}(j')^\rho)} \Big[ \|z_{i''}^{L-2} - z_{j''}^{L-2}\| \Big]}_{\text{\textcircled{1'}}} \\
& \underbrace{+ 2 K_\phi^{(L-1)} K_\phi^{(L-2)} \sum_{i'' \in \mathcal{N}(i')^\rho, j'' \in \mathcal{N}(j')^\rho} T_{i'',j''}^{(L-2,\mathcal{N}(i')^\rho \mathcal{N}(j')^\rho)} \Big[ \|z_{i''}^{L-3} - z_{j''}^{L-3}\| \Big]}_{\text{\textcircled{2'}}} \\
& \cdots \\
& \underbrace{+ (L-1) K_\phi^{(L-1)} K_\phi^{(L-2)} \cdots K_\phi^{(1)} \sum_{i'' \in \mathcal{N}(i')^\rho, j'' \in \mathcal{N}(j')^\rho} T_{i'',j''}^{(L-2,\mathcal{N}(i')^\rho \mathcal{N}(j')^\rho)} \Big[ \|x_{i''} - x_{j''}\| \Big] \Big]}_{\text{\textcircled{$(L-1)'$}}} \Bigg]
\end{aligned}
$$

Now we see something familiar: $\text{\textcircled{1'}}$ to $\text{\textcircled{$(L-1)'$}}$ can all be bounded in the same way, e.g.,

$$
\begin{aligned}
\text{\textcircled{1'}} \leq K_\phi^{(L-1)} \sum_{i'' \in \mathcal{N}(i')^\rho, j'' \in \mathcal{N}(j')^\rho} T_{i'',j''}^{(L-2,\mathcal{N}(i')^\rho \mathcal{N}(j')^\rho)} \Bigg[ & K_\phi^{(L-2)} K_\phi^{(L-3)} \cdots K_\phi^{(1)} \|x_{i'} - x_{j'}\| + \\
& K_\phi^{(L-2)} \sum_{i''' \in \mathcal{N}(i'')^\rho, j''' \in \mathcal{N}(j'')^\rho} T_{i''',j'''}^{(L-3,\mathcal{N}(i'')^\rho \mathcal{N}(j'')^\rho)} \Big[ \|z_{i''}^{L-3} - z_{j''}^{L-3}\| \Big] \\
& K_\phi^{(L-2)} K_\phi^{(L-3)} \sum_{i''' \in \mathcal{N}(i'')^\rho, j''' \in \mathcal{N}(j'')^\rho} T_{i''',j'''}^{(L-3,\mathcal{N}(i'')^\rho \mathcal{N}(j'')^\rho)} \Big[ \|z_{i''}^{L-4} - z_{j''}^{L-4}\| \Big] \\
& \cdots \\
& K_\phi^{(L-2)} K_\phi^{(L-3)} \cdots K_\phi^{(1)} \sum_{i''' \in \mathcal{N}(i'')^\rho, j''' \in \mathcal{N}(j'')^\rho} T_{i''',j'''}^{(L-3,\mathcal{N}(i'')^\rho \mathcal{N}(j'')^\rho)} \Big[ \|x_{i'''} - x_{j'''}\| \Big] \Bigg] := bound(\text{\textcircled{1'}})
\end{aligned}
$$

Similarly, plugging the bound gives

$$
\begin{aligned}
= & \sum_{i \in V_a^\rho, j \in V_b^\rho} T_{i,j}^{(L, V_a V_b)}\left[\left(\prod_{m=1}^{L} K_\phi^{(m)}\right)\|x^i - x^j\| \right.\\
& + K_\phi^{(l)} \sum_{i' \in \mathcal{N}(i)^\rho, j' \in \mathcal{N}(j)^\rho} T_{i',j'}^{(L-1, \mathcal{N}(i)^\rho \mathcal{N}(j)^\rho)}\left[L \cdot K_\phi^{(L-1)} K_\phi^{(L-2)} \cdots K_\phi^{(1)} \|x_{i'} - x_{j'}\| \right.\\
& + K_\phi^{(l-2)} \sum_{i'' \in \mathcal{N}(i')^\rho, j'' \in \mathcal{N}(j')^\rho} T_{i'',j''}^{(L-2, \mathcal{N}(i')^\rho \mathcal{N}(j')^\rho)}\left[(1+2+\cdots+(L-1)) \cdot K_\phi^{(L-2)} K_\phi^{(L-3)} \cdots K_\phi^{(1)} \|x_{i'} - x_{j'}\| \right.\\
& + 1 \cdot K_\phi^{(l-2)} \sum_{i''' \in \mathcal{N}(i'')^\rho, j''' \in \mathcal{N}(j'')^\rho} T_{i''',j'''}^{(L-3, \mathcal{N}(i'')^\rho \mathcal{N}(j'')^\rho)}\left[\|z_{i''}^{L-3} - z_{j''}^{L-3}\|\right]\\
& + (1+2) \cdot K_\phi^{(l-2)} K_\phi^{(l-3)} \sum_{i''' \in \mathcal{N}(i'')^\rho, j''' \in \mathcal{N}(j'')^\rho} T_{i''',j'''}^{(L-3, \mathcal{N}(i'')^\rho \mathcal{N}(j'')^\rho)}\left[\|z_{i''}^{L-4} - z_{j''}^{L-4}\|\right]\\
& \cdots \\
& \left.\left.\left.+ (1+2+\cdots(l-2)) \cdot K_\phi^{(l-2)} K_\phi^{(l-3)} \cdots K_\phi^{(1)} \sum_{i''' \in \mathcal{N}(i'')^\rho, j''' \in \mathcal{N}(j'')^\rho} T_{i''',j'''}^{(L-3, \mathcal{N}(i'')^\rho \mathcal{N}(j'')^\rho)}\left[\|x_{i'''} - x_{j'''}\|\right]\right]\right]\right]
\end{aligned}
$$

We can see that the weight of the center nodes gradually changes as: $L \to 1 + 2 + \cdots (L-1) \to 1 + (1+2) + \cdot(1+2+\cdots+(L-2)) \to \cdots$. These are exactly the elements at level $L+1$ of the Pascal's triangle. See Figure 3 for an illustration. For instance, if $L = 4$, the $L+1$ level of Pascal's triangle is $(1, 4, 6, 4, 1) = (1, 1+(1+2), (1+2+3), 4, 1)$, which matches the weight of center nodes in each level.

Let $P_L^l$ is the $l$-th number at level $L$ of Pascal's triangle. Applying the bound recursively and extracting $\prod_{m=1}^{L} K_\phi^{(m)}$ gives the following bound:

$$
\begin{aligned}
& \Delta_{T^{(L, V_a^\rho V_b^\rho)}}^{(L)} \\
\leq & \left(\prod_{m=1}^{L} K_\phi^{(m)}\right) \sum_{i \in V_a^\rho, j \in V_b^\rho} T_{i,j}^{(L, V_a V_b)}\left[\|x^i - x^j\| + \frac{L}{1} \cdot \sum_{i' \in \mathcal{N}(i)^\rho, j' \in \mathcal{N}(j)^\rho} T_{i',j'}^{(L-1, \mathcal{N}(i)^\rho \mathcal{N}(j)^\rho)}\left[\|x_{i'} - x_{j'}\| \right.\right.\\
& \left.\left.+ \frac{1+2+\cdots+(L-1)}{L} \cdot \sum_{i'' \in \mathcal{N}(i')^\rho, j'' \in \mathcal{N}(j')^\rho} T_{i'',j''}^{(L-2, \mathcal{N}(i')^\rho \mathcal{N}(j')^\rho)}\|x_{i'} - x_{j'}\| + \cdots\right]\right]\\
= & \left(\prod_{m=1}^{L} K_\phi^{(m)}\right) \sum_{i \in V_a^\rho, j \in V_b^\rho} T_{i,j}^{(L, V_a V_b)}\left[\|x^i - x^j\| + \frac{P_{L+1}^L}{P_{L+1}^L} \cdot \sum_{i' \in \mathcal{N}(i)^\rho, j' \in \mathcal{N}(j)^\rho} T_{i',j'}^{(L-1, \mathcal{N}(i)^\rho \mathcal{N}(j)^\rho)}\left[\|x_{i'} - x_{j'}\| \right.\right.\\
& \left.\left.+ \frac{P_{L+1}^{L-1}}{P_{L+1}^L} \cdot \sum_{i'' \in \mathcal{N}(i')^\rho, j'' \in \mathcal{N}(j')^\rho} T_{i'',j''}^{(L-2, \mathcal{N}(i')^\rho \mathcal{N}(j')^\rho)}\|x_{i'} - x_{j'}\| + \cdots\right]\right]
\end{aligned}
$$

By setting each transportation plan to the optimal plan acquired from the TMD OT problem, the bound is equivalent the tree mover distance with depth-$L+1$ computation tree, and $w(l) = P_{L+1}^{l-1}/P_{L+1}^l$. The one with $\epsilon \neq 1$ can be trivially extended from the current proof. □

## A.5 Tree Mover's Distance Stability Proof

### A.5.1 Node Drop

*Proof.* We use a illustration to provide the conceptual idea of the proof.

Given a graph $G = (V, E)$, let $G'$ be the graph where node $v \in V$ is dropped. The computation tree of node $v$ will play an important role here. Firstly, the nodes in level $l$ of the computation tree are the nodes that can reach $v$ within $l-1$ steps. In particular, the number of nodes in level $l$ determines how

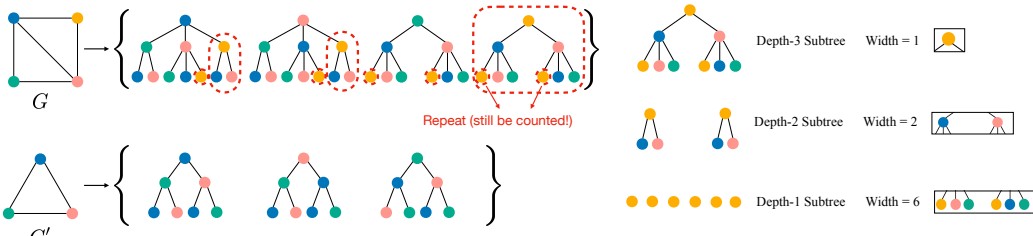

Figure 8: **Illustration of Node Drop.**

many depth-$(L - l + 1)$ computation tree of $v$ exists in $\mathcal{T}_G$. Therefore, deleting $v$ from the graph will also delete all the subtrees in $\mathcal{T}_G$ that are rooted at $v$, and the number of deleted trees are determined by the width of $T_v$. See Figure 8 for a illustration. Deleting a subtree $T$ will introduce an additional transportation cost $\text{TD}_w(T, T_\emptyset)$, which is the tree norm of subtree. By aggregating all the tree norms and consider the effect of weights $w$, we arrive at

$$\text{TMD}_w^L(G, G') \leq \sum_{l=1}^{L} \lambda_l \cdot \underbrace{\text{Width}_l(T_v^L)}_{\text{Tree Size}} \cdot \underbrace{\text{TD}_w(T_v^{L-l+1}, T_\emptyset)}_{\text{Tree Norm}},$$

where $\text{Width}_l(T)$ is the width of $l$-th level of tree $T$ and $\lambda_1 = 1$, $\lambda_l = \prod_{j=1}^{l-1} w(L + 1 - j)$. This is an upper bound instead of equality as some deleted subtrees are repeated counted in the bound. $\square$

### A.5.2   Edge Drop

*Proof.* We again use a illustration to provide the conceptual idea of the proof.

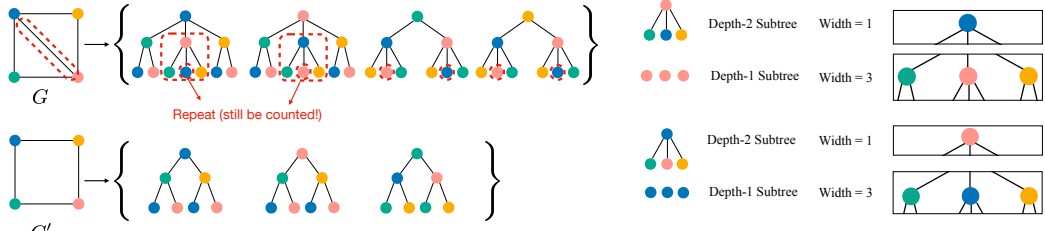

Figure 9: **Illustration of Edge Drop.**

Given a graph $G = (V, E)$, let $G'$ be the graph where edge $(u, v) \in E$ is dropped. Different from node drop, deleting an edge will affect both nodes. In particular, the number of nodes in level $l$ of computation tree $T_v$ determines how many depth-$L - l$ computation tree of $u$ exists in $\mathcal{T}_G$. Therefore, deleting edge $u - v$ from the graph will also delete all the subtrees in $\mathcal{T}_G$ that are rooted at $v$ ($u$) where the roots have ancestor $u$ ($v$). See Figure 10 for a illustration. Similarly, by aggregating all the tree norms and consider the effect of weights $w$, we arrive at

$$\text{TMD}_w^L(G, G') \leq \sum_{l=1}^{L-1} \lambda_{l+1} \cdot \left( \text{Width}_l(T_v^L) \cdot \text{TD}_w(T_u^{L-l}, T_\emptyset) + \text{Width}_l(T_u^L) \cdot \text{TD}_w(T_v^{L-l}, T_\emptyset) \right).$$

where $\text{Width}_l(T)$ is the width of $l$-th level of tree $T$ and $\lambda_1 = 1$, $\lambda_l = \prod_{j=1}^{l-1} w(L + 1 - j)$. This is an upper bound instead of equality as some deleted subtrees are repeated counted in the bound. $\square$

### A.5.3   Node Perturbation

*Proof.* The node perturbation is the simplified case of node drop, where only the node features are perturbed.

Given a graph $G = (V, E)$, let $G'$ be the graph where node feature $x_v$ is perturbed to $x_v'$. The tree mover's distance between $G$ and $G'$ is equal to

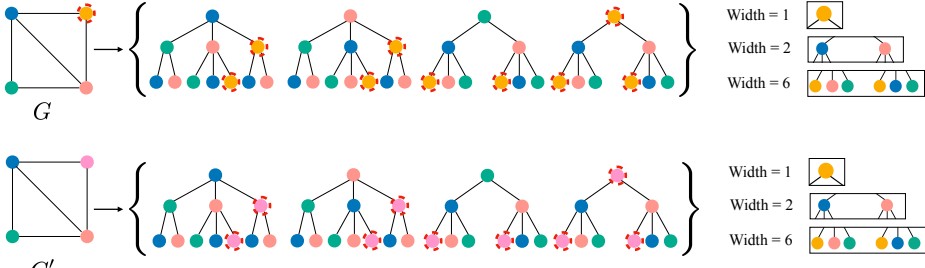

Figure 10: **Illustration of Node Perturbation.**

$$\text{TMD}_w^L(G, G') \le \sum_{l=1}^{L} \lambda_l \cdot \text{Width}_l(T_v^L) \cdot \left\| x_v - x_v' \right\|.$$

where $\text{Width}_l(T)$ is the width of $l$-th level of tree $T$ and $\lambda_1 = 1$, $\lambda_l = \prod_{j=1}^{l-1} w(L + 1 - j)$. This is an upper bound instead of equality as changing node features would affect the optimal coupling of OT. We might not have the optimal transportation plan after node perturbation. $\qquad\square$

## B  Lipschitz Condition for Other GNNs

### B.1  Graph Convolutional Network

Here, we consider Graph Convolutional Network (GCN) [27] with the following message passing rules:

$$\begin{array}{ll} \substack{\text{Message} \\ \text{Passing}} & z_v^{(l)} = \phi^{(l)} \left( z_v^{(l-1)} + \epsilon \mathbb{E}_{u \in \mathcal{N}(v)} z_u^{(l-1)} \right), \end{array} \quad \begin{array}{ll} \substack{\text{Graph} \\ \text{Readout}} & h(G) = \phi^{(L+1)} \left( \mathbb{E}_{u \in V} z_u^{(L)} \right). \end{array}$$

In particular, the SUM is replaced with MEAN in the message passing and graph readout. It is easy to show that the same bound holds by replacing all the unnormalized OT with normalized OT* in tree distance and tree mover's distance, i.e.,

$$\text{TD}_w^*(T_a, T_b) = \begin{cases} \left\| x_{r_a} - x_{r_b} \right\| + w(L) \cdot \text{OT}_{\text{TD}_w}^*(\rho(\mathcal{T}_{r_a}, \mathcal{T}_{r_b})) & \text{if } L > 1 \\ \left\| x_{r_a} - x_{r_b} \right\| & \text{otherwise}, \end{cases}$$

$$\text{TMD}_w^{*L}(G_a, G_b) = \text{OT}_{\text{TD}_w}^*(\rho(\mathcal{T}_{G_a}^L, \mathcal{T}_{G_b}^L)).$$

This gives the bound:

$$\left\| \text{GCN}(G_a) - \text{GCN}(G_b) \right\| \le \prod_{l=1}^{L+1} K_\phi^{(l)} \cdot \text{TMD}_w^{*L+1}(G_a, G_b),$$

### B.2  Other Message Passing GNNs

Sometimes, the center nodes are treated differently compared to the neighbors, e.g.,

$$\begin{array}{ll} \substack{\text{Message} \\ \text{Passing}} & z_v^{(l)} = \phi^{(l)} \left( z_v^{(l-1)} + \varphi^{(l)} \left( \sum_{u \in \mathcal{N}(v)} z_u^{(l-1)} \right) \right), \end{array} \quad \begin{array}{ll} \substack{\text{Graph} \\ \text{Readout}} & h(G) = \phi^{(L+1)} \left( \sum_{u \in V} z_u^{(L)} \right) \end{array}$$

The proof can be easily modified as follows:

$$\left\| \phi^{(l)}\left( z_i^{(l-1)} + \varphi^{(l)}\left( \sum_{i' \in \mathcal{N}(i)} z_i^{(l-1)} \right) \right) - \phi^{(l)}\left( z_j^{(l-1)} + \varphi^{(l)}\left( \sum_{j' \in \mathcal{N}(i)} z_j^{l-1} \right) \right) \right\|$$

$$\leq K_\phi^{(l)} \left\| \left( z_i^{(l-1)} + \varphi^{(l)}\left( \sum_{i' \in \mathcal{N}(i)} z_i^{(l-1)} \right) \right) - \left( z_j^{(l-1)} + \varphi^{(l)}\left( \sum_{j' \in \mathcal{N}(i)} z_j^{(l-1)} \right) \right) \right\|$$

$$\leq K_\phi^{(l)} \left( \left\| z_i^{(l-1)} - z_j^{(l-1)} \right\| + \left\| \varphi^{(l)}\left( \sum_{i' \in \mathcal{N}(i)} z_i^{(l-1)} \right) - \varphi^{(l)}\left( \sum_{j' \in \mathcal{N}(i)} z_j^{(l-1)} \right) \right\| \right)$$

$$\leq K_\phi^{(l)} \left( \left\| z_i^{(l-1)} - z_j^{(l-1)} \right\| + K_\varphi^{(l)} \left\| \sum_{i' \in \mathcal{N}(i)} z_i^{(l-1)} - \sum_{j' \in \mathcal{N}(i)} z_j^{(l-1)} \right\| \right)$$

Therefore, simply replacing the $\epsilon$ with $K_\varphi^{(l)}$ leads to a Lipschitz bound for this variant:

$$\left\| h(G_a) - h(G_b) \right\| \leq \prod_{l=1}^{L+1} K_\phi^{(l)} \cdot \text{TMD}_w^{L+1}(G_a, G_b),$$

where $w(l) = K_\varphi^{(l-1)} \cdot P_{L+1}^{l-1}/P_{L+1}^l$ for all $l \leq L$ and $P_L^l$ is the $l$-th number at level $L$ of Pascal's triangle.

## C  Additional Experiments

### C.1  Graph Clustering

One advantage of graph metric over GNNs is that we can perform geometric analysis of graph datasets such as graph clustering. We adopt k-medoids [24], a variant of k-means [30], to perform unsupervised clustering with TMD. Figure 11 provides an qualitative example of clustering, and Table 3 measures the quallity of clusters with Normalized Mutual Information (NMI) and Completeness Score (CS) [6]. We can see that k-medoids with tree mover's distance generates meaningful clusters that aligns with labels.

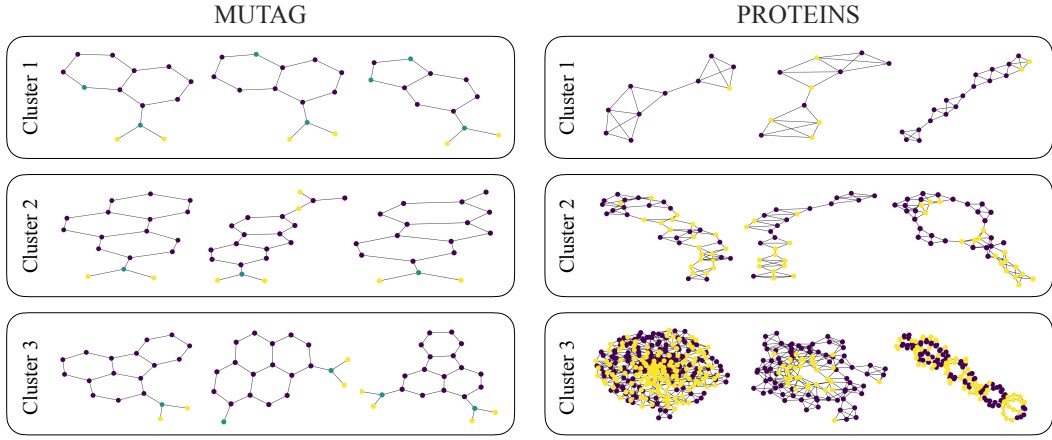

Figure 11: **Unsupervised Clustering with TMD.** Node attributes are indicated by colors.

### C.2  t-SNE Visualization of Graphs

Equipped with TMD, we can extend t-SNE [48] from Euclidean space to graphs. Specifically, t-SNE constructs a probability distribution based on *pairwise distance* and minimizes the divergence between

|  | MUTAG (K=2) | | PROTEINS (K=2) | | ENZYMES (K=6) | |
|---|---|---|---|---|---|---|
|  | NMI | CS | NMI | CS | NMI | CS |
| TMD L=1 | 25.6±8.1 | 25.0±7.9 | 6.58±0.56 | 7.26±0.61 | 6.55±0.81 | 6.73±0.74 |
| TMD L=2 | **30.4±8.9** | **29.7±8.9** | 7.70±0.90 | 8.24±0.69 | **6.70±1.11** | **6.90±1.01** |
| TMD L=3 | 28.9±7.3 | 28.0±7.2 | 8.28±0.67 | 8.76±0.87 | 6.53±0.65 | 6.69±0.64 |
| TMD L=4 | 26.6±5.4 | 25.8±5.4 | **9.22±0.01** | **9.91±0.78** | 6.34±0.60 | 6.57±0.55 |

Table 3: **Unsupervised Clustering on TU Dataset.** The number of clusters K is equal to the number of graph classes. The performance is measured by Normalized Mutual Information (NMI) and Completeness Score (CS).

the distribution of low dimensional and original data. We simply replace the Euclidean distance in conventional t-SNE with tree mover's distance and show the t-SNE visualization of various graph datasets in Figure 12 . Although not perfectly, we can observe the separation between points with different labels.

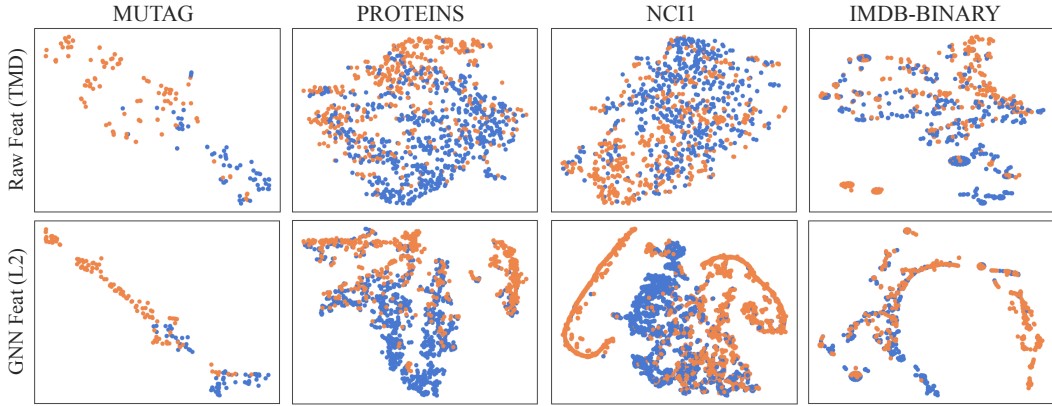

Figure 12: **t-SNE Visualization of Graph Datasets.** The (binary) classes are indicated by colors. The upper row shows the t-SNE visualization of input space with TMD and the bottome rows shows the representation space of trained GNNs with Euclidean distance.

## C.3 Additional Results for Section 5

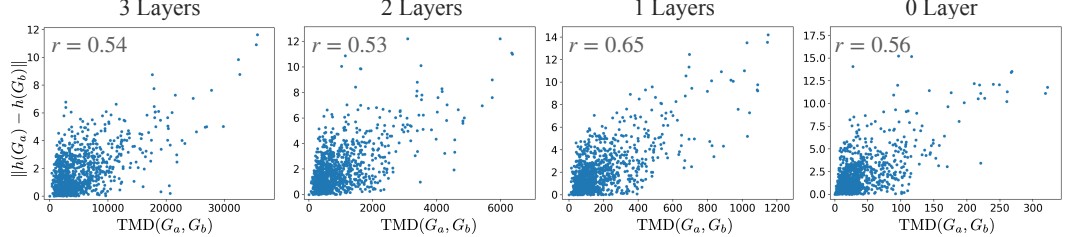

Figure 13: **Correlation between GNNs and TMD on PROTEINS.** The Pearson correlation coefficient $r$ between $\|h(G_a) - h(G_b)\|$ and TMD / WWL are showed on the upper left of the figures.

We repeat the experiments in section 5 with PROTEINS dataset and show the results in Figure 13 and 14. We can see that the GNNs are less sensitive to small graph perturbation as Figure 14 shows, as the number of nodes and edges is much larger than the one in MUTAG.

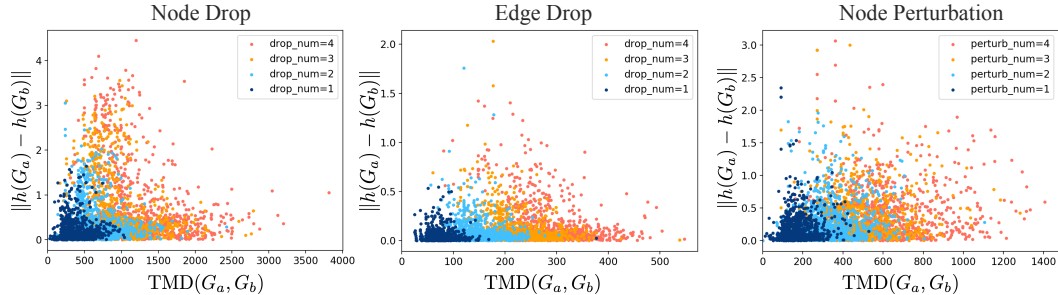

Figure 14: **Robustness under Graph Perturbation on PROTEINS.**