# OpenReview forum: "Tree Mover's Distance: Bridging Graph Metrics and Stability of Graph Neural Networks"
_NeurIPS.cc/2022/Conference — NeurIPS 2022 Accept_

### Official Review · Reviewer_QSky · 2022-07-08

**Rating:** 6
**Confidence:** 4
**Soundness:** 3 good
**Presentation:** 3 good
**Contribution:** 4 excellent

**Summary:**

The authors first propose TMD, a pseudometric for comparing graphs to each other. TMD compares graphs to each other by recursively solving a series of optimal transport problems which minimizes distances of subtree patterns. The proposed pseudometric is evaluated in graph classification (distances fed to an indefinite kernel and then to SVM). The results indicate that TMD performs on par with the best performing baselines. The authors also provide some theoretical results. First, they bound the Lipschitz constant of the GIN model with respect to TMD, and also analyze the stability of GIN under node deletions, edge deletions and perturbations of node features. Finally, they provide a result about the generalization error of GIN under distribution shifts.

**Questions:**

For the graph classification task, it is not clear to me why an SVM classifier with indefinite kernel is employed. Since TMD computes distances, the natural way to perform classification would be to employ the k-nn classifier. Did the authors experiment with that classifier and how does it compare to the SVM classifier?

Typos:\
l.205: "the empirically performance" --> "the empirical performance"


**Limitations:**

The authors discuss the limitations and potential negative societal impact of their work.

**Strengths And Weaknesses:**

Strengths:
- In my view, the originality of the paper is high. The proposed TMD distance is novel. Previous studies have applied optimal transport techniques on the labels produced by the WL algorithm, but in my understanding, the proposed recursive definition is different from previous work.

- I really like the results about the stability of graph neural networks. Most previous studies have focused on a different problem: whether a graph neural network can distinguish classes of non-isomorphic graphs or not. Not much work has been done on the distance between the graph representations. I am not thought sure how useful the experiments of subsection 5.3 are. What's the purpose of showing that the correlation between TMD and a graph neural network is high? Furthemore, does this hold for all datasets?

Weaknesses:
- The computational complexity of the proposed distance function is very high since it needs to solve a series of optimal transport problems. This renders the method practically infeasible for datasets that contain large graphs (such as REDDIT-BINARY) and datasets that contain many samples (such as the OGB graph property prediction datasets). Of course, for the optimal transport problem, one could use some approximate algorithm, but still I don't think the proposed method can be applied to large datasets.

- As one can see in Table 1, the proposed method provides only marginal improvements in accuracy over the baselines. Furthermore, TMD is not compared against several state-of-the-art graph neural networks. It is only compared against GIN and GCN which are at most as expressive as 1-WL. It would be nice if the authors could also report the running time of TMD and compare it against that of the Wasserstein WL function.

- The message passing scheme of GIN given in subsection 5.1 is different from the one provided in the original paper, i.e., $z_v^{(l)} = \phi^{(l)} ((1+\epsilon^{(l)})z_v^{(l-1)} +\sum_{u \in \mathcal{N}(v)} z_u^{(l-1)})$. Furthermore, both for the experiments and for the proof of Theorem 8, the authors set $\epsilon=1$. GIN is known to be less expressive than 1-WL when $\epsilon=1$. Furthermore, the message passing scheme of GCN given in Appendix B.1 is not correct. Thus, I wonder whether the Lipschitz constant of any message passing graph neural network can be bounded under TMD or are there some conditions that need to be satisfied?

- Even though TMD is sufficiently different from the Wasserstein WL pseudometric, I would suggest the authors provide more details about how the two methods differ from each other and also compare the complexities of the two approaches to each other.

---

> ### Author Response · Authors · 2022-08-02
> **Response to Reviewer QSky**
>
> Thank you for your constructive and helpful suggestions. We would like to address your questions as follows:
>
> 1. Computation Complexity of TMD: As stated in section 4.2, the runtime of TMD is highly dependent on the node degrees. Fortunately, the pairwise distance between subtrees can be calculated in parallel for each depth, which can benefit from parallel programming  for large graphs. Here, we state the average computational time of calculating pairwise distances with WWL and TMD over 100 random sampled pairs on the DD and NCI1 datasets, where DD contains large graphs (AVG #Node: 284.32, AVG #Edge: 715.66) and NCI1 contains small graphs (AVG #Node:  29.87, AVG #Edge: 32.30). The parallel version of TMD is executed with 3 processes.
>
> [DD] WWL: 7.92 sec/pair; TMD: 32.07 sec/pair; TMD_Parallel: 24.44 sec/pair
>
> [NCI1] WWL: 0.11 sec/pair; TMD: 0.34 sec/pair; TMD_Parallel: 0.81 sec/pair
>
> Compared to WWL, the runtime of TMD_Parallel is roughly three times larger than WWL, which is much faster than the (worst case) theoretical Big-O analysis.
>
> 2. Classification Benchmark: The goal of TMD is not outperforming SOTA graph neural networks, e.g., those that are provably more powerful than 1-WL. Instead, TMD serves as a new framework to study the stability and generalization of GNNs. Nevertheless, compared to DS-GNN-ED proposed in table 1 of [1], which is a new GNN that is strictly more powerful than 3-WL, TMD still outperforms it on MUTAG, PTC, and NCI1 dataset. For running time of TMD and WWL, please refer to the response above.
>
> 3. Formulation of GIN: First, the GIN paper weighs the center node with $\epsilon$, while we weigh the neighbors. Second, GIN’s $\epsilon$ is layer dependent while we fix the $\epsilon$ as a constant. The purpose of these changes is to simplify the notation of TMD. One can easily derive an equivalent form by changing the weight function $w$ of TMD to recover the original formulation of GIN. We adopted a variant of the GCN formulation from Xu et al. [1], where mean is adopted instead of sum. The stability of most message passing GNNs can be bounded by TMD, but involves a more complicated weight function.
>
> 4. Comparison to WWL: The WWL is simply a Wasserstein distance between distributions over node features. Therefore, its expressiveness highly relies on the node features, whereas TMD is well-defined without any external feature extractor. In addition, TMD provably bounds the stability of GNNs, which is very useful for understanding the robustness and generalization of GNNs.
>
> 5. Results of KNN: The table below shows the mean over 10 independent trials with 90%/10% train-test split with kNN classifier. We can see that kNN classifiers with TMD perform similarly to SVM classifiers. The reason to adopt SVM classifier was to follow the evaluation scheme of FGW.
>
> |      | MUTAG (K=1) |  PTC_MR (K=10) | PROTEINS (K=10) |  NCI1 (K=10) |
> | ----------- | ----------- | ----------- | ----------- |  ----------- |
> | L=0      | 83.8       | 63.8 | 73.0 | 67.7 |
> | L=1   | 88.3        | 65.9 | 74.1 | 76.6 |
> | L=2   | 93.9        | 66.8 | 74.3 | 78.7 |
> | L=3   | 93.9        | 61.4 | 75.7 | 80.0 |

---

> > ### Comment · Reviewer_QSky · 2022-08-09
> > **Response to Authors**
> >
> > I would like to thank the authors for the clarifications provided in their response. I understand that the main focus of the paper is not on proposing a state-of-the-art approach for graph classification, however the authors need to somehow validate that TMD is a credible distance function of graphs. Since all theoretical results for GNNs depend on the proposed function, TMD being a meaningful function would make GNNs more attractive. This can be verified experimentally by conducting further experiments. In my view, TMD's empirical performance is not weak, but I would like to see some more experiments (also in tasks different from graph classification).

---

> > > ### Author Response · Authors · 2022-08-09
> > > **Thank you for your response**
> > >
> > > Thank you for your response! In the appendix, we also demonstrate the applications of TMD in graph clustering (C.1) and t-SNE visualization of graphs (C.2). Without supervision, TMD can still capture meaningful structure of graphs. In summarization, we demonstrate the applicability of TMD in terms of graph classification, clustering, and t-SNE visualization while showing that TMD can estimate the stability and generalization of GNNs.

---

### Official Review · Reviewer_DmtC · 2022-07-11

**Rating:** 5
**Confidence:** 4
**Soundness:** 2 fair
**Presentation:** 3 good
**Contribution:** 2 fair

**Summary:**

This paper introduces a graph pseudo-metric based on the hierarchical Optimal Transport , to understand the generalization of machine learning models on graphs. They show that the proposed TMD captures properties relevant to graph classification and can be related to generalization of GNNs under distribution shifts.

**Questions:**

1. Can the authors provide an example regarding “the distance $TMDLw(Ga, Gb)$ can be zero even if $Ga = Gb$”.
2.  The notation in Definition 4 is confusing. What is the $OT_{TD_w}$?
3. The complexity of iteratively calculating the distance between the L-depth trees is high. The running time of baselines and the proposed methods shall be added.


**Limitations:**

Yes

**Strengths And Weaknesses:**

Strengths:

1. The paper is well-written and easy to follow, although there exist some unclear descriptions.
2. This paper proposes a new OT distance for graphs. Using the computation trees of the graph to calculate the distance between two graphs is direct and reasonable.
3. The Lipschitz Constant and Stability analysis of GNNs seem to be useful and meaningful.

Weaknesses:

1. ”TMD can provably distinguish graphs that are identifiable by the (1-dimensional) Weisfeiler-Leman graph isomorphism test”. The authors say it can be further strengthened by augmenting node attributes e.g. with positional encodings, but they have not provided the details. Since expressive power is very important for graph representation learning.
2.  The computational complexity of TMD is high. The authors only implement it on CPU (POT package). It is unclear whether the method can be accelerated by GPUs.
3. Some baselines on graph OT are missing, for example, [1] [2] bellow.


[1] GOT: An Optimal Transport framework for Graph comparison
[2] COPT: Coordinated Optimal Transport on Graph.

---

> ### Author Response · Authors · 2022-08-02
> **Response to Reviewer DmtC**
>
> Thank you for your constructive and helpful suggestions. We would like to address your questions as follows:
>
> 1. Augmenting node attributes: This strengthening is analogous to that of message passing GNNs with positional encodings [3][4]. In particular, by augmenting additional node attributes, the representation power and performance of GNNs can be improved [3]. Similarly, the representation power of TMD can also be strengthened if it uses additional node attributes, e.g., positional encodings.
> Overall, the goal of this paper is to provide a framework that studies the stability and generalization with new graph metrics instead of improving the graph representation learning. We will clarify this to prevent confusion.
>
> [3] Dwivedi et al., Graph Neural Networks with Learnable Structural and Positional Representations, ICLR 2022
>
> [4] You et al., Position-aware Graph Neural Networks, ICML 2019
>
>
> 2. Computational Complexity: The computational complexity mainly comes from the computation of OT. This computation, especially in the hierarchical form, is not well amenable to speedups from GPUs, as it cannot easily be written as matrix multiplications. Instead, here we demonstrate that parallel programming can speed up TMD for large graphs. In particular, the pairwise distance between subtrees can be calculated parallely for each depth, which can benefit from parallel programming  if the graphs are large.
>
> In particular, here we show the average computation time of calculating pairwise distance with Wasserstein Weisfeiler-Lehman (WWL) kernels and TMD over 100 random sampled pairs on the DD and NCI1 datasets, where DD contains large graphs (AVG #Node: 284.32, AVG #Edge: 715.66) and NCI1 contains small graphs (AVG #Node:  29.87, AVG #Edge: 32.30). The parallel version of TMD is executed with 3 processes.
>
> [DD] WWL: 7.92 sec/pair; TMD: 32.07 sec/pair; TMD_Parallel: 24.44 sec/pair
>
> [NCI1] WWL: 0.11 sec/pair; TMD: 0.34 sec/pair; TMD_Parallel: 0.81 sec/pair
>
> Hence, and not unexpected, parallelization improves the computation time of TMD on large graphs, while its improvement on small graphs is limited.
>
> 3. Baseline on Graph OT: Thank you for pointing this out. [1] focuses on transportation of graph signals derived from graph Laplacians and [2] is more similar to the FGW baseline: both of them leverage Gromov-Wasserstein to capture the structural properties of graphs. A major difference between TMD and these previous works is that the goal of TMD is to understand the stability and generalization of GNNs instead of outperforming SOTA graph kernels or GNNs on classification benchmarks.
>
> 4. Examples of where $\textnormal{TMD}(G_a,G_b)$ can be zero:  Any graph that cannot be distinguished by 1-WL serves as an example. For instance, see Figure 1,2 in [5].
>
> [5] Vikas et al., Generalization and Representational Limits of Graph Neural Networks, ICML2020
>
> 5. Definition 4: It is the OT distance defined in (2) with $\textnormal{TD}_w$ as the metric.
>
> 6. Runtime comparison: We will include a table to compare the runtime between baselines in the final version. For preliminary experiments, please refer to the response 2.
>
> To conclude, we want to re-highlight our unique contributions. Without the recursive OT formulation, it is non-trivial to construct a proper distance that precisely bound the stability of GNNs. Despite being computationally more expensive, it opens a new path to study the stability and generalization of GNNs, as the Lipschitz condition is one of the most important components in learning theory. Thank you again for your suggestions, we hope that our clarifications help the reviewer in reassessing the paper.

---

> > ### Comment · Reviewer_DmtC · 2022-08-08
> > **Thanks a lot for your reply!**
> >
> > After reading the reply from the authors, I think some of my concerns are not well addressed.  For example, regrading baselines on graph OT,  you have stated that TMD works good for a classification problem (in the abstract), then why not compare the performance of TMD to a range of methods for graph classification?  You present TMD as a pseudometric, then it is natural to compare it with other graph metrics.
> > Meanwhile, I would highly encourage the authors to take into account of the comments and revise the manu. accordingly.

---

> > > ### Author Response · Authors · 2022-08-09
> > > **Thank you for your response**
> > >
> > > Thank you for your response! Due to page limits, we only compare TMD with closely related baselines. Here, we provide a more comprehensive comparison to some of the latest graph metrics and kernels:
> > >
> > > |      | MUTAG |PTC |PROTEINS |NCI1 |NCI109 |BZR |COX2 |
> > > | ----------- | ----------- |----------- |----------- |----------- |----------- |----------- |----------- |
> > > | TMD | 92.2 | 68.5 | 75.2 | 84.8 | 82.8 | 85.5 | 79.1 |
> > > | FGOT [1] | 86.78 | - | 66.22 | - | - | 82.07 | - |
> > > | SG-OT [2] | 88.72 | 64.58 | - | - |  69.57 | 85.65 |  80.15 |
> > > | HGK-WL [3] | - | - | 75.93 | - |  - | 78.59 |   78.13 |
> > > | GH [4] | - | - | 74.78 | - |  - | 76.49 |  76.41 |
> > > | P-WL-UC [5] | 85.17 | 63.46 | 75.86 | 85.62 |   85.11 | - |  - |
> > > | DeepGK [6] | 82.66 | 57.32 | 71.68 | 62.48 |   62.69 | - |  - |
> > > | RetGK [7] | 90.3 | 62.5 | 75.8 | 84.5 |  - | - | - |
> > > | WLS [8] | - | - | - | - |  - | 84.64 |   79.07 |
> > >
> > >
> > > We can see that TMD still outperforms or performs competitively with the baselines. We will include them in the paper once the page limit is increased.

---

### Official Review · Reviewer_pkBn · 2022-07-11

**Rating:** 7
**Confidence:** 3
**Soundness:** 4 excellent
**Presentation:** 3 good
**Contribution:** 4 excellent

**Summary:**

A metric on the set of graphs is defined using concepts from optimal transport.  It is shown both analytically and by experiment that graph neural networks define Lipschitz continuous functions in the metric.


**Questions:**

Let me check in an example that I understand how the definition works for graphs without features.
Let T_N be the tree with a root and N children.  The distance TD(T_m,T_n)=|n-m| because
the distance between a blank tree (feature vector = 0) and a generic vertex (feature vector = 1) is 1.

If so then for graphs with features, the definition depends on the point 0 chosen for the blank tree feature.
This should be mentioned more explicitly.

**Limitations:**

Technical results with no immediate societal impact.

**Strengths And Weaknesses:**

Strength: The proposal is clear and well motivated, and has evident applications.  The evaluation is adequate for a first work.

Weakness: Graph metrics are a much studied field and there is little comparison with previous work.

---

> ### Author Response · Authors · 2022-08-02
> **Response to Reviewer pkBn**
>
> Thank you for your positive and encouraging evaluation. For attributed graphs, we simply use the zero vector as the point 0. This is inspired by the vector norm, where the norm of a vector is its distance from zero vector. We extend this concept to structured objects such as tree and graph by augmenting blank nodes that have zero vectors as features. Due to page limits, we primarily compare TMD with optimal transport / WL based graph kernels. More discussion will be provided in the final version.

---

### Official Review · Reviewer_tWYt · 2022-07-13

**Rating:** 6
**Confidence:** 4
**Soundness:** 3 good
**Presentation:** 3 good
**Contribution:** 3 good

**Summary:**

Authors introduce a new (pseudo) distance on attributed graphs, the Tree Mover's distance (TMD). They first introduce Then authors introduce a distance between trees (TD) which aims at recursively comparing their roots, through their respective attributes, and their subtrees using optimal transport (OT) to get hard assignments between these induced subtrees to pursue the recursion i.e comparing trees of smaller respective depth. TMD then naturally comes from TD by modeling graphs as multisets of trees rooted in each node of each graph. TMD is shown to define a proper pseudo-metric for which the axiom of discernability is closely related to the Weisfeiler-Lehman graph isomorphism test. After investigating the revelance of TMD for graphs classification, authors further study its relevance to quantify stability and generalization abilities of the well-known Graph Isomorphism Networks (GIN), being SOTA graph neural networks.

**Questions:**

For clarity and conciseness my questions/doubts have been clearly stated in the "Strengths and weaknesses" section.



**Limitations:**

A few limitations of their work have not really been addressed as illustrated by my elaboration on the "weaknesses/points for clarification" paragraph. The authors have adequately addressed the potential negative societal impact of their work in the supplemental material.

**Strengths And Weaknesses:**

$\textbf{Strengths}$:

- Overall the paper is well-written and the design of TMD is elegant/original. The authors have managed to clearly address a wide variety of concepts, from kernels to GNNs. An obvious pedagogical effort has been made in several proofs of the results provided, which is appreciable.

- TMD is interesting by its flexibility suggested by its dependencies to a depth-dependent weighting function and to cost functions inherent to used OT (on nodes).

- TMD seems competitive as a kernel in graphs classification and mostly shines on the study of GIN.

$\textbf{Neutral about}$:

- Without a clear characterization of TMD balls in the space of attributed graphs (or at first, unattributed graphs), I find it difficult to envision how TMD can guide GNN to further improvements, but given the difficulty of this task, the empirical evidence provided in Sections 5 and 6 supports the relevance of TMD for this purpose.

$\textbf{Weaknesses / points to clarify}$:

- Authors exploit specific properties of the Kantorovich formulation of OT (especially its relation to Monge's formulation) which are eluded in the paper and clearly not straightforward so, to improve the clarity of the document e.g the need for definitions 2 and 3, it would be good to mention them (Also the caption of figure 1 can be improved for this purpose).

- Theorem 7: From your proof I would say that the stated implication is actually an equivalence. Could you elaborate on this ?

- On the graphs classification benchmark : I am not sure to understand your validation scheme from your explanations. From my understanding you did a cross-validation for TMD, while e.g authors of FGW reported a 10-fold nested cross-validation in their paper  (Which better quantifies generalization abilities and is more natural for graph kernel methods as the computational bottleneck lies in the computation of the kernel matrix). Therefore I suggest harmonizing the validation scheme on kernel methods instead of just reporting the performance of the respective papers. Moreover, could you complete the benchmark on graphs classification with a benchmark in terms of runtimes ?

- On subsection 5.1: There is a difference between your formulation of message-passing and the one from GIN (see equation 4.1 of [48]), $\epsilon$ is not handled in the same way. As you set $\epsilon=1$ for your experiments they are still valid, but the implications of this change for the theoretical results in your paper and the ones in GIN's paper are not clear to me, even if it seems minor. Could you elaborate on this ?

- There is no reference to the figure 4 and 5 in the main paper.

**Modification:** I increased my initial grade from 5 (borderline accept) to 6 (weak accept) after a convincing rebuttal and discussion by the authors.

---

> ### Author Response · Authors · 2022-08-02
> **Response to Reviewer tWYt**
>
> Thank you for your constructive and helpful suggestions. We would like to address your questions as follows:
>
> 1. Clarification of OT formulation: We will provide more discussion w.r.t. Unbalanced OT (Definition 2 and 3) and its relationship to Monge’s formulation.
>
> 2. Discriminative power of TMD: Yes, the other direction can also be proved. Sketch: If TMD > 0, the computation trees will be different. The WL test is essentially comparing sets of computation trees, so WL also agrees that these two graphs are different. We only present one direction to simplify the presentation.
>
> 3. Graph Classification Benchmark: Sorry for the confusion. Analogous to FGW, we also report a 10-fold nested cross-validation. We will include a table to compare the runtime between baselines in the final version. For preliminary experiments, here we compare the runtime between Weisfeiler-Lehman (WWL) kernels, TMD, and a parallel version of TMD on the DD and NCI1 datasets, where DD contains large graphs (AVG #Node: 284.32, AVG #Edge: 715.66) and NCI1 contains small graphs (AVG #Node:  29.87, AVG #Edge: 32.30). The parallel version of TMD is executed with 3 processes.
>
> [DD] WWL: 7.92 sec/pair; TMD: 32.07 sec/pair; TMD_Parallel: 24.44 sec/pair
>
> [NCI1] WWL: 0.11 sec/pair; TMD: 0.34 sec/pair; TMD_Parallel: 0.81 sec/pair
>
> Compared to WWL, the runtime of TMD_Parallel is roughly three times larger than WWL, which is much faster than the (worst case) theoretical Big-O analysis.
>
>
> 4. Difference w.r.t. GIN: First, the GIN paper weighs the center node with $\epsilon$, while we have a weight on the neighbors. Second, GIN’s $\epsilon$ is layer dependent, while we fix the $\epsilon$ as a constant. The purpose of these changes is to simplify the notation of TMD. One can easily derive an equivalent form by changing the weight function $w$ of TMD to recover the original formulation of GIN.
>
> 5. Reference to Figure 4 and 5: They belong to Section 5.3. We will fix this in the final version.
>
>
> To conclude, we would like to highlight our unique contributions. Without the recursive OT formulation, it is non-trivial to construct a proper distance that precisely bounds the stability of GNNs. Despite being computationally more expensive, it opens a new path to study the stability and generalization of GNNs, as the Lipschitz condition is one of the most important components in learning theory. Thank you again for your suggestions, we hope that our clarifications help the reviewer in reassessing the paper.

---

> > ### Comment · Reviewer_tWYt · 2022-08-08
> > **Response to rebuttal**
> >
> > Thank you for the clarifications provided in your concise rebuttal. I still have some concerns and few additional comments on the paper.
> >
> > 1. In my opinion, the clarification/correction of the discussions around the OT distances in the paper is mandatory: In practice OT as defined in (1) or (2), can also be defined between any pair of empirical distributions {x_i | i <= n} and {y_j | j <=  m} with variable number of samples (i.e, n different from m), as long as these samples are in the same vectorial space (e.g R^p) and the total mass of each distribution coincides (total mass of 1 in the paper). Authors focus on a definition between pair of distributions with the same number of samples and design TMD based on it. The question WHY is not well adressed in my opinion in the paper, could you clarify this point ?
> >
> > L112-113: The considerations around unbalanced and partial OT are wrong imo. For instance, the unbalanced OT [8] aims at comparing arbitrary nonnegative Radon measures where total mass of the first distribution may vary from the total mass of the second. It does not have anything to do with the number of samples in each distribution as stated by the authors: “To compute the OT between sets with different sizes …”
> >
> > 2. Why do not all the combinations of TD between subtrees (4 combinations) appear ? If there are considerations made over the OT plan between both multisets of subtrees (line 1 of the figure), they should be detailed.
> >
> > 3. Lemma 13 leading to Theorem 6 is wrong as it is, even if it is easy to fix: The optimal coupling (let's say T) between X and Y is not an admissible coupling for the padded versions of X and Y. So the first argument provided L532-533 is wrong. But it does hold if you consider, for instance the padding of T with an identity matrix on the rows/columns assigned to the 0s.
> >
> > 4. On the graph clustering experiments provided in the supplementary material (subsection C.1): I believe that you did not aim at exactly computing Fréchet means (barycenters) w.r.t TMD (nor FGW), but instead got an approximated solution of the centroids by following neirest neighbor heuristics on a pre-computed pairwise distance matrix. Do you confirm ? If it is the case, it should be written in the subsection. Otherwise, the optimization/algorithm to estimate Fréchet means w.r.t TMD should be detailed. Overall, a spectral clustering fixing the number of cluster and the number of components to the true number of clusters would have been more appropriate considering the difficulty (np-hard) to compute Fréchet means with these two distances.  Considering your classification benchmark, which is convincing, it is not a major flow of the paper.
> >
> > 5. Why L+1 in equation (3) of the main paper instead of L ?
> >
> > tipos/minor flows:
> > - L40: “via on a” -> “via a”
> > - L89: “The graph may have with node features” -> “The graph may have node features”
> > - L297-298: “based on number of nodes” → “based on the number of nodes”
> >
> > - L109: the notion of equality between trees is used but not introduced before. Such issue is redundant either between trees or graphs in the paper, a comment should be made somewhere for the sake of clarity.
> >
> >
> > Sorry for the delay in my response but I would like to get the detailed explanations (especially) on the point 1 above from the authors to know if I am raising my grade or not.

---

> > > ### Author Response · Authors · 2022-08-09
> > > **Thank you for your response**
> > >
> > > Thank you for your response!
> > >
> > > 1. The reason for using augmentations is to increase the discriminative power. For instance, consider the following example: there are three graphs with no edges, graph_a has nodes {x, x}, graph_b has nodes {y, y}, graph_c has nodes {y, y, y, y}. For standard OT, the distances of pairs (graph_a, graph_b) and (graph_a, graph_c) are the same as the mass is normalized to 1. Nevertheless, with blank augmentation, our OT can distinguish this case. WIth regard to the reference of unbalanced OT, our formulation is mainly inspired by [7] instead of [8]. We will further clarify the difference between unbalanced OT approaches in the final version.
> > >
> > > 2. There are only two combinations in the second row of Figure 1 as, for simplicity of the illustration, we assume there exists a unique optimal OT solution. Therefore, the optimal solution is either (blue tree, yellow tree), (green tree, blank tree) or (green tree, yellow tree), (blue tree, blank tree). We simply present the first case in Figure 1. We will explicitly state this and clarify that we are only presenting the optimal solution by assuming that blue tree and yellow tree are more similar (lower cost).
> > >
> > > 3. Sorry for the confusion. Padding T with an identity matrix for zero vectors is indeed what we would like to describe in Lemma 13. We omit this as costs between zero vectors are zero, which will not affect the overall cost. We will add these details in the proof.
> > >
> > > 4. As L671 describes, we adopt k-medoids as an alternative of k-means. In particular, we adopt the Partitioning Around Medoids (PAM), where the algorithm does not compute a “mean”, but only relies on the pairwise distance to perform clustering.
> > >
> > > 5. This comes from Thm 5, where for L-layer GNNs, the TMD bound computes distances between (L+1)-depth trees.
> > >
> > > Thank you again for pointing out the typos. We will modify them and improve the clarity in the final version.

---

### Meta-Review · Area_Chair_oMGe · 2022-08-23

**Recommendation:** Accept
**Confidence:** Certain

**Metareview:**

This paper proposes a new similarity measure between graphs, based on computing optimal transport between distributions of trees extracted from graphs. The method benefit from the fast solvers of OT between trees and the proposed metric has been shown to be interesting for computing  a "Lipshitz" constant related to the generalization of message passing GNN. The experiments were appreciated but lack of comparison with existing graph distances and GNN was noted by the reviewers on the graph classification experiment.

The authors did a very good reply to the reviewers which was much appreciated. For instance the new experiments are very interesting and should be included in the paper or supplementary. The fact that the performance does not depend too much on the classifier (SVM VS KNN) is also interesting. During discussion the consensus was that the paper deserves to be published at NeurIPS but that the authors are requested to include the new results and discussions/clarifications in the paper and supp.

**Award:**

No

---

### Decision · Program_Chairs · 2022-09-14

Accept